# Global Merger-Arbitrage Forecasting with Language Models

Hinal Jajal [1]    Michal Mucha [1]    Charles Sweat [1]    Chris Pulman [1]    Charlie Flanagan [1]    Peter Anderson [1]

## Abstract

We present a language-model forecasting system for merger arbitrage, a specialized high-stakes financial setting in which the task is to predict the outcome of announced M&A deals. Unlike prior work on judgmental forecasting with LLMs, which has focused on broad mixed-topic benchmarks and short context such as news snippets, we study a setting that requires long-context reasoning over hundreds of pages of technical documents. Our system combines expert-guided context engineering with finetuning on hindsight-guided reasoning traces derived from historical deals. Given an announced deal, it outputs a probability distribution over three mutually exclusive outcomes: closing at announced terms, a higher bid, or deal termination. On an out-of-sample set of more than 400 large deals spanning 42 countries, our finetuned system achieves the best performance of any method we evaluate, reducing class-balanced Brier score to 0.151. This is 24% below calibrated market-implied probabilities, 19% below XGBoost, and 25-42% below frontier language models. These results, together with ablation studies, show that LLM-based forecasting can succeed in specialized, long-context financial workflows, with hindsight-based supervision and expert-designed context playing a critical role.

## 1 Introduction

Judgmental forecasting—probabilistic prediction of future events based on documents and contextual reasoning, rather than explicit statistical models—is increasingly being studied in the context of large language models. Recent work shows that LLMs can produce calibrated probability forecasts approaching the accuracy of some human forecasters (Halawi et al., 2024; Schoenegger et al., 2024; Alur et al., 2025). However, this literature largely focuses on broad, mixed-topic question banks, posed with only shallow context such as news headlines and blurbs (Karger et al., 2025). It remains unclear whether LLM-based forecasting can add value in specialized, high-stakes domains that require analyzing long and highly technical texts.

We focus on a financial domain: forecasting the outcomes of announced mergers and acquisitions (M&A). Such forecasts are actively monetized in merger-arbitrage strategies, where an investor buys the target firm's stock after a deal is announced (possibly against a short position in the acquirer) to capture the spread between the current price and the deal consideration. The investor earns this spread if the deal closes and faces losses if the deal terminates. Therefore, accurate probability forecasts of deal closure and alternative resolutions are central to position sizing, risk management, and portfolio construction.

Arriving at these probability forecasts requires processing large volumes of complex text. Analyzing a single deal can require reading a hundred-page merger agreement; assessing competitive overlaps, regulatory regimes and political considerations across jurisdictions to anticipate regulatory responses; comparing against outcomes in similar past transactions; scrutinizing shareholder bases, voting histories, and public statements; reviewing risks in the acquirer's financing and balance sheet, and more. At the same time, analysts must integrate this information with a continuous stream of regulatory filings, press releases, and expert analyses.

We build an LLM-based forecasting system tailored to this setting. First, retrieval-augmented research agents each gather and analyze a specific dimension of deal context. Second, an ensembled, finetuned frontier model uses the resulting in-depth context (typically 6.8-10k tokens) to generate a probabilistic forecast and a detailed deal report. The research agents are optimized over months of collaboration with veteran merger-arb specialists. The frontier model is finetuned on gold training targets derived from post-mortem analysis. For each historical deal, we generate hindsight-guided forecasts at multiple points in time to identify the evidence that should have been weighted most heavily given the realized outcome. We enforce strict temporal integrity, for example by excluding open-web sources prone to date-

[1]Balyasny Asset Management. Correspondence to: Hinal Jajal <bamappliedai@bamfunds.com>.

*Proceedings of the $43^{rd}$ International Conference on Machine Learning*, Seoul, South Korea. PMLR 306, 2026. Copyright 2026 by the author(s).

filter leakage, and use only models with knowledge cutoffs preceding the test period. This process yields quality reasoning traces that train the model to reason about deal risk and generate meaningful reports.

On a held-out test set of 404 large deals spanning 42 countries, our finetuned system outperforms frontier models, probabilities inferred from market prices, and an XGBoost model trained on a rich set of deal features across a range of weighted Brier scores. Its best class-balanced Brier score is 0.151, compared with 0.199 for Platt-scaled market-implied probabilities, 0.186 for XGBoost, and 0.201–0.259 for frontier models given identical context (lower is better). Finetuning drives much of this gain, reducing calibration error from 0.089 to 0.039. More broadly, LLM-based approaches are less tied to market consensus than XGBoost: the correlation with market-implied probabilities is 0.36 for our finetuned system, compared with 0.75 for XGBoost.

Ablation studies show that each component contributes materially to performance: class-balanced Brier score worsens by 0.032 without hindsight guidance, by 0.021 when the system is limited to Deal Card information, and by 0.026 when training on only half of the 1244-deal training set. On the basis of these results, we deploy the system as a decision-support tool for analysts and portfolio managers.

## 2 Related Work

Halawi et al. (2024) are the first to demonstrate that LLMs, when provided with retrieved news context, can approach non-expert human performance in backtests on broad prediction-market question sets—especially early in the forecast, when human forecasters are uncertain and at least five relevant news articles are available. Building on this, Hsieh et al. (2024) augment LLM forecasters with a ReAct (Yao et al., 2022) loop that incorporates web search and Python tools. In a complementary direction, Schoenegger et al. (2025) show that LLM assistants improve human forecasting accuracy.

More recently, AIA Forecaster (Alur et al., 2025) focuses on agentic search, forecast calibration, and ensembling with a supervisor agent, matching expert human performance on ForecastBench. They also introduce MarketLiquid, constructed from liquid prediction-market contracts, and show that while their system alone underperforms the market-implied probabilities, an ensemble that combines AIA Forecaster with market prices yields forecasts that are more accurate than market prices alone.

Given the many potential sources of information leakage in backtesting LLM forecasters (Paleka et al., 2025), recent work has focused on standardizing and improving evaluation protocols. Sudhir et al. (2024) propose a suite of logical consistency checks. Bench to the Future (Wildman et al.,

2025) mitigates contamination by providing a per-question snapshotted web corpus. To address the erosion of static benchmarks as LLM knowledge cutoffs advance, Forecast-Bench (Karger et al., 2025) continuously aggregates unresolved questions from prediction platforms and prediction markets, enabling evaluation of humans and LLMs. Dai et al. (2025) introduce a continuous evaluation that uses daily news as an oracle, automatically generating forecasting questions and showing that accuracy degrades smoothly as evaluation dates drift beyond training cutoffs, even with retrieval-augmented generation.

Complementing these backtests, small-scale live evaluations have been reported. Schoenegger et al. (2024) evaluate 12 LLMs on 31 prediction market questions within 48 hours of question opening, finding no statistical difference in performance between the human crowd and the LLM crowd. Alur et al. (2025) also report a small-scale live evaluation that is competitive with prediction markets.

Across all evaluation protocols—static, dynamic, and live—prior work has converged on mixed sets of heterogeneous questions. We argue that investigating *only broad, mixed-topic question banks* is a mistake. Combining sports, geopolitics, climate, macroeconomics, politics, and more collapses qualitatively different forecasting problems into a single score, obscuring where models succeed or fail, providing poor guidance for model and system development, and incentivizing shallow, generic retrieval and prompting. This setup also systematically ignores the domain-specific data, tools, and workflows that define specialist forecasting in practice, and therefore risks overstating the readiness of current LLM forecasters for deployment.

In contrast to previous work, we study a single, economically important domain in depth and design our system around the artifacts and workflows of specialist forecasters. Rather than treating retrieval as generic web search over news snippets, we build a retrieval and reasoning pipeline targeted at long, technical financial documents. Methodologically, aside from Halawi et al. (2024) and Turtel et al. (2025), there is little work on finetuning to improve judgmental forecasts, and thus limited understanding of how to construct gold targets. We address this gap and demonstrate the value of finetuning hindsight into the model.

## 3 Task and Dataset

### 3.1 Task Definition

We aim to predict the outcome of an announced merger or acquisition. The prediction has three components:

**1. Deal Outcome Probabilities.** We distinguish between three mutually exclusive outcomes:

1. **Succeed$^+$**: The deal closes as announced.
2. **Fail$^+$**: The announced deal is terminated, but the target shareholders experience a *positive* outcome as a higher bid emerges.
3. **Fail$^-$**: The announced deal is terminated with a *negative* outcome for target shareholders (no higher bid).

Differentiating between positive and negative terminated deals is crucial for commercializing predictions, and for comparing to market-implied probability (refer Sec. 3.2.1).

**2. Days to Completion.**  Prediction of the days to deal completion, assuming the deal closes as announced.

**3. Deal Report.**  A detailed explanation of the reasoning underlying the predicted probability distribution, including citations and key deal-specific risks and mitigants to monitor, such as a termination fee or a pre-existing lawsuit. Identifying key risks in the deal is central to merger-arb, especially for discretionary—rather than systematic—traders.

### 3.2   Dataset Construction

There is no readily available dataset of historical M&A deals and their outcomes. We construct a historical sample of 1,648 public-target M&A deals covering the 4 year period from 1 January 2022 to 31 December 2025. This horizon balances the need for a sufficiently deep history to support finetuning against the difficulty of reconstructing rich deal context from older text sources.

We exclude transactions that do not represent classic, control-oriented merger-arb situations—specifically, minority-stake acquisitions ($\leq 50\%$), asset sales, and SPAC transactions—where there is no clean, tradable takeover spread for target shareholders. We also exclude small deals (e.g. <US\$1bn deal value) that lack institutional liquidity.

Several key fields in the original dataset are incomplete. We use an ensemble of LLM-based ReAct (Yao et al., 2022) agents, with access to the search tools defined in Sec. 4.1.1, to enrich each deal with the following metadata:

- Announcement date and resolution date.
- Deal terms and their revision history (e.g., consideration type, currency, cash per share, exchange ratio).
- Company guidance on expected closing timing and subsequent updates.
- Presence and sequence of competing bids.

Incomplete or ambiguous information is reviewed manually.

#### 3.2.1   MARKET-IMPLIED PROBABILITY

After deal announcement, the target's stock price incorporates investors' expectations about deal resolution, allowing us to infer the *market-implied probability* of a favorable outcome for target shareholders. We use this both as a training signal for the model and as an input to baselines for comparison on out-of-sample deals.

Let $S_t$ denote the target's share price on day $t$ after the announcement. Investors' valuation is approximated as a two-state mixture between:

- an upside value $S_t^+$ corresponding to a positive outcome (**Succeed$^+$** or **Fail$^+$**), and
- a downside value $S_t^-$ corresponding to a negative outcome (**Fail$^-$**).

Under this model,

$$S_t \approx p_t^m S_t^+ + (1 - p_t^m) S_t^-,$$

where $p_t^m$ is the market-implied probability of a positive outcome. This two-state mixture representation of the target's payoff is standard in derivative and event-driven valuation; see, for example, Hull (2012). Solving for $p_t^m$ yields

$$p_t^m = \text{clamp}_{[0,1]} \left( \frac{S_t - S_t^-}{S_t^+ - S_t^-} \right)$$

For cash deals, $S_t^+$ is constant and equal to the per-share cash consideration, discounted by the risk-free rate over the expected time-to-close inferred from company guidance. When no company guidance is available, we use the median days-to-close for U.S. public deals: 175 days. For stock or mixed deals, $S_t^+$ is the per-share value implied by the announced terms at time $t$ (e.g., exchange ratio times the acquirer's share price, plus the discounted value of any cash component). The value $S_t^+$ reflects the prevailing deal terms at time $t$, and any amendments (e.g. an increase in the offer price) result in an adjustment to $S_t^+$.

We model the downside $S_t^-$, the expected target price if the deal fails, by allowing standalone value to evolve with market movements scaled by the target's pre-announcement beta $\beta$:

$$S_t^- = S_0^- \exp\big(\beta \, r_{m,[0,t]}\big),$$

where $S_0^-$ is the 20-day pre-announcement average price and $r_{m,[0,t]}$ is the cumulative log return of the market index from announcement to day $t$. While this specification captures the effect of prevailing market conditions, realized post-break prices also depend on the reason for termination and the potential for alternative transactions, and therefore vary substantially in practice. Consequently, market-implied probability is informative but should not be treated as definitive; in fact, it benefits from further calibration through Platt scaling (refer Sec. 5).

*Table 1.* Dataset splits, outcome distribution, and total forecast instances across time.

| Split | Deal Outcomes (%) | | | Total Deals | Total Forecasts | Time Period |
|---|---|---|---|---|---|---|
| | Succeed$^+$ | Fail$^+$ | Fail$^-$ | # | # | |
| Train | 84.4 | 2.2 | 13.4 | 848 | 2032 | Jan 2022 – Jan 2024 |
| Validation | 82.8 | 3.6 | 13.6 | 396 | 1008 | Feb 2024 – Jan 2025 |
| Test | 84.0 | 3.0 | 14.4 | 404 | 1115 | Feb 2025 – Dec 2025 |
| **Overall** | 84.0 | 2.7 | 13.7 | 1648 | 4155 | Jan 2022 – Dec 2025 |

### 3.2.2 FORECAST DATES AND TEMPORAL SPLITS

We partition deals into training, validation, and test sets based on announcement date, using temporal cutoffs of *31 January 2024* and *31 January 2025*. For each deal, we generate multiple forecast dates between the announcement date and the company-guided expected close date. If no guided close date is available, we instead sample forecast dates at two-month intervals up to the actual close date. All forecast dates are constrained to lie within the temporal bounds of their respective data split.

We use the validation set for hyperparameter selection. For final model evaluation, we retrain on the union of the training and validation sets and report performance on the held-out test set. Table 1 summarizes the data splits, including the number of deals, forecast instances, and outcomes.

### 3.3 Information Leakage

Paleka et al. (2025) highlight several leakage mechanisms that can contaminate LLM forecasting backtests. Our setting mitigates many of these issues.

**Model knowledge cutoffs.** We only evaluate LLMs whose knowledge cutoff precedes 31 January 2025 (the start of the test period). The same constraint applies to the embedding and reranking models used for retrieval.

**System-prompt leakage.** Hidden vendor prompts can leak contemporaneous facts (e.g., the current U.S. president), contaminating forecasting benchmarks. In our domain, such prompts are unlikely to reveal deal-specific outcomes.

**Piggybacking on human forecasts.** LLMs may copy human forecasts leaked or deliberately admitted into retrieval context, thereby achieving the appearance of near-human performance. In our setting, both expert commentary and documents that mention stock prices or deal spreads can appear in context, which can give the model an implicit read on market-implied probabilities. This is not, by itself, a flaw: our primary baselines rely on market-implied probability directly. In Sec. 5 we show that our system significantly outperforms market implied probability and is less correlated with the market than the XGBoost baseline.

**Benchmark gaming via correlated questions.** Generic question banks can include many correlated risks that enable gaming (Sempere & Lawsen, 2021). In contrast, M&A outcomes are largely idiosyncratic at the deal level.

**Question selection.** Backtests can leak information through question selection (e.g., conditioning on resolution by a known evaluation date, or biasing towards events that have actually occurred). We reduce this risk by constructing the universe mechanistically from public-target deals.

**Date-restricted retrieval.** Enforcing temporal cutoffs is notoriously difficult with web search or news APIs: pages are continuously edited, news stories are updated, and search providers inject or scrape real-time widgets that bypass nominal date filters (Alur et al., 2025). We avoid this class of leakage entirely by explicitly excluding news feeds and open-web content. Instead, we restrict context to curated and timestamped regulatory filings, company releases, and other commercial data sources (refer Sec. 4.1.1).

**Hindsight-guided finetuning.** Section 4.2.1 describes our approach to finetuning on hindsight-guided deal reports. The finetuned model is exposed to postmortems, realized outcomes, and future evidence only for training-set deals, never for test-set deals. Accordingly, hindsight-based supervision cannot inflate test-set performance.

### 3.4 Evaluation Metrics

We predict a distribution over the realized outcome $Y \in \{\textbf{Succeed}^+, \textbf{Fail}^+, \textbf{Fail}^-\}$. For evaluation, we consider two binary tasks derived from $Y$: (i) **Succeed/Fail** (deal completes vs. terminates), and (ii) $+/-$ (positive vs. negative outcome for shareholders). The $+/-$ task can be evaluated against market-implied probabilities (Section 3.2.1).

Predicted probabilities over deal outcomes are evaluated using the Brier score (Brier, 1950), a proper scoring rule that incentivizes calibration. The standard Brier score—mean squared error applied to probabilistic forecasts—does not account for class imbalance ($\approx 84\%$ of deals complete) or the asymmetric risk of merger-arb. We therefore report four Brier scores: the standard score **Brier**, a class-balanced variant **Brier**$_B$ (Pratt et al., 2024), a surprise-weighted variant **Brier**$_S$, and a P&L-weighted variant **Brier**$_\$$.

**Weighted Brier Scores.** Let $Z_t \in \{0,1\}$ be the binary indicator for the relevant task (e.g., $Z_t = \mathbf{1}\{Y = \textbf{Succeed}^+\}$ for **Succeed/Fail**, or $Z_t = \mathbf{1}\{Y \in \{\textbf{Succeed}^+, \textbf{Fail}^+\}\}$ for $+/-$). For a forecast $\hat{p}_t \in [0,1]$ of the corresponding positive class, all weighted variants share the form

$$\textbf{Brier}_W = \mathbb{E}_w\big[(Z_t - \hat{p}_t)^2\big]$$

where $\mathbb{E}_w$ denotes a weighted expectation. Weights $w$ are:

- **Class-balanced** ($\textbf{Brier}_B$): inverse class frequency, equalizing total weight on each class.
- **Surprise-weighted** ($\textbf{Brier}_S$): $w = 1 - p_t^m$ for successes ($Y = \textbf{Succeed}$), $w = p_t^m$ for failures ($Y = \textbf{Fail}$), emphasizing outcomes the market misread.
- **P&L-weighted** ($\textbf{Brier}_\$$): $w = \frac{S_t^+ - S_t}{S_t}$ for successes and $w = \frac{S_t - S_t^-}{S_t}$ for failures, emphasizing the available return for a correct prediction.

All weighted variants normalize weights so each class contributes equally (50%) and clip outlier weights to prevent dominance by extreme cases. We report scores averaged over forecast dates within each deal, then across deals.

**Murphy Decomposition.** LLM forecasts are often poorly calibrated (Schoenegger et al., 2024; Alur et al., 2025). We apply the Murphy decomposition (Murphy, 1973) to separate calibration error from resolution. This allows us to distinguish models that assign systematically biased probabilities from those that simply fail to rank deals by risk.

**Market Alignment.** We evaluate how closely forecasts $\hat{p}_t$ track market-implied probabilities $p_t^m$ via mean squared deviation $(\hat{p}_t - p_t^m)^2$ and Pearson correlation $\rho(\hat{p}_t, p_t^m)$. These distinguish models that replicate market assessments from those introducing systematic deviations.

**Days to completion.** The number of days to deal completion is evaluated using *mean absolute percentage error* (**MAPE**) with respect to the realized number of days, a standard scale-free accuracy measure (Flores, 1986).

## 4 Approach

### 4.1 Architecture

Illustrated in Fig. 1, our approach comprises 12 tool-augmented ReAct (Yao et al., 2022) agents, each responsible for gathering and analyzing a specific dimension of deal context, together with a forecasting module based on an ensembled, finetuned LLM that synthesizes probabilistic forecasts and a final report. This two-stage design separates *information retrieval* (research agents) from *probabilistic reasoning* (forecasting module), facilitating independent optimization of each stage while ensuring interpretable in-

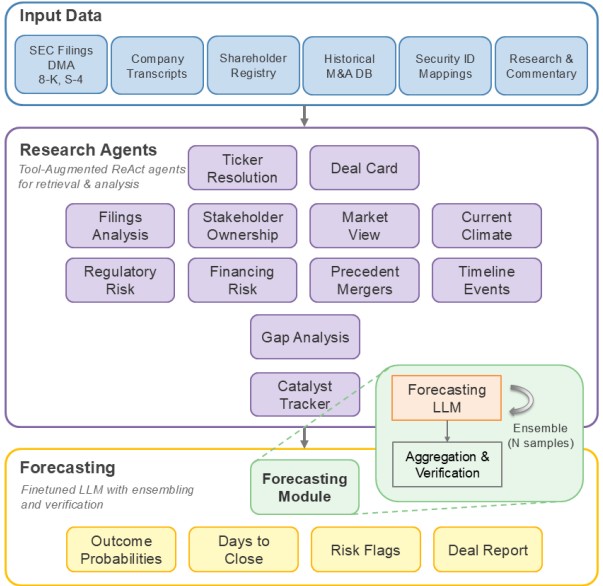

*Figure 1.* System architecture combining 12 specialized, ReAct research agents with an ensembled, fine-tuned forecasting LLM.

termediate representations.

#### 4.1.1 INPUT DATA AND RESEARCH AGENTS

Input data is obtained from six sources:

1. **Regulatory Filings**: Definitive merger agreements (DMA), proxy statements, S-4 and 8-K filings, etc.
2. **Company Transcripts**: Transcripts of company earnings calls and presentations.
3. **Shareholder Registry**: Point-in-time institutional stock ownership data.
4. **Historical M&A Database**: Records of past merger attempts, outcomes, and timelines.
5. **Security ID Mappings**: Identifiers linking entities and tickers across data systems.
6. **Research & Commentary**: Expert analysis and other research.

Documents are stored in a large-scale vector store built on a domain-finetuned embedding model (Anderson et al., 2024), together with a finetuned reranker for passage-level relevance. The indexed corpus is intentionally broad rather than restricted to the deals in our evaluation set: it spans global public and private companies, sectors, commodities, macroeconomic developments, and geopolitical events, and comprises millions of documents. The same retrieval stack underlies interactive tools used by human analysts and portfolio managers (and their AI assistants), so agents operate on the same document distribution and interfaces that support real-world forecasting workflows.

*Table 2.* Research agents and their functions in execution order (Phases 1–4). Each agent collects and analyzes specific contextual information about the M&A transaction, which is subsequently integrated by the forecasting agent.

| Phase | Research Agent | Responsibilities |
|---|---|---|
| 1 | Ticker Resolution | Maps company names to unique identifiers across multiple financial data systems (security IDs, entity IDs, organization IDs), enabling cross-platform data retrieval for both acquirer and target companies. |
| | Deal Card | Extracts structured deal metadata from SEC filings and expert commentary, including parties, structure (merger type, consideration components), key dates, regulatory requirements, termination fees, and voting thresholds. |
| 2 | Filings Analysis | Analyzes SEC filings (merger agreements, proxy statements, 8-Ks, S-4s) to extract key facts, identify red flags (adverse terms, litigation risks) and green flags (deal protection provisions), and flag unusual or notable clauses. |
| | Stakeholder Ownership | Retrieves and analyzes major shareholder positions ($\geq$5% and $\geq$10% thresholds) for both acquirer and target, identifying potential voting dynamics, activist involvement, ownership concentration, and alignment of incentives. |
| | Market View | Reviews expert commentary and market sentiment regarding the deal, examining pre-deal health of both parties, industry trends, rumors, and opinions on deal rationale and synergies. |
| | Current Climate | Assesses the broader market, regulatory, and economic environment relevant to the transaction, including antitrust climate, interest rate environment, sector-specific headwinds or tailwinds, and geopolitical factors. |
| | Regulatory Risk | Analyzes antitrust and regulatory approval risks, international regulatory requirements, industry concentration concerns, and historical precedent for similar deals. |
| | Financing Risk | Assesses risks related to deal financing and acquirer's ability to complete the transaction, including debt financing availability, equity raise requirements, credit rating implications, covenant compliance, and acquirer financial health. |
| | Precedent Mergers | Identifies and analyzes similar historical M&A transactions from deal databases, filtering by industry, geography, deal size, and time period. Ranks precedents by relevance and extracts completion rates, timelines, and key risk factors. |
| | Timeline Events | Extracts and organizes key events in the deal lifecycle (regulatory filings, shareholder votes, court dates, financing milestones), categorizes by type, and identifies past, ongoing, and future expected catalysts. |
| 3 | Gap Analysis | Performs gap analysis across all previously collected module outputs to identify missing information or overlooked details relevant to deal completion probability, ensuring comprehensive coverage. |
| 4 | Catalyst Tracker | Identifies forward-looking catalysts and near-term developments that could materially impact deal probability or timeline (upcoming regulatory decisions, shareholder vote dates, financing deadlines, competing bid windows). |

All documents are timestamped. Retrieval is constrained to items released prior to the forecast date to prevent temporal leakage (see Sec. 3.3). Agents interact with this corpus through tools that expose full-text search, retrieval, and structured lookups (e.g., holdings, identifiers, and deal attributes). Each research agent is implemented using GPT-4.1 or GPT-5 with access to a subset of these tools and is prompted to extract domain-specific information from retrieved context.

Agent roles, prompt formulations, and output schemas were iteratively optimized over months of collaboration with veteran merger-arb specialists, who reviewed system outputs on historical deals they had previously covered, compared the automated analysis to their own research, and highlighted gaps for correction. The agents, their responsibilities, and their execution order are summarized in Tab. 2.

### 4.1.2 FORECASTING MODULE

The forecasting module uses a designated *forecast LLM* (a finetuned model, or a frontier model such as GPT-5) to convert the research agents' outputs into probabilistic predictions and a final deal report.

**Context assembly.** We first concatenate all agent outputs into a single structured context with citations, and pair this with a forecasting prompt that instructs the forecast LLM to start from historical base rates (completion frequencies), in-

corporate deal-specific evidence, and explicitly weigh green flags against red flags. The resulting context is typically from 6.8k-10k tokens (interquartile range), with a median of 8.3k tokens.

**Ensemble generation.** The forecast LLM is then conditioned on this context and sampled $N = 5$ times with temperature $0.2$ to produce an ensemble of independent forecasts. Each sample includes outcome probabilities, a forecast of days to completion, and a draft deal report.

**Aggregation & verification.** For robustness to outliers, we take the median of the $N$ samples as the final prediction for both outcome probabilities and days to completion; Alur et al. (2025) show that the median of five samples recovers most of the benefit of larger ensembles. To obtain the final deal report, we provide GPT-5 (with tool access) the set of sampled reports. It cross-checks inconsistencies against source documents and produces a single, consolidated, and fact-checked report.

### 4.2 Finetuning

We finetune the *forecast LLM* on historical deals with known outcomes, allowing it to learn patterns that drive merger success or failure and to map qualitative evidence to calibrated forecasts. A central design question is how to construct gold training targets for: (i) the deal report, which is the model's reasoning chain, and (ii) quantitative outputs (probabilities

over outcomes and days to completion).

Recent work on reasoning-focused LLMs emphasizes that naive supervision on final outcomes alone is insufficient: models benefit from targets that reflect high-quality *process* as well as correct *answers*, often via teacher distillation and process supervision (Uesato et al., 2022; Lightman et al., 2024; Luo et al., 2024). Our setting is unusual and arguably more favorable: for each forecast date, we observe the full future evolution of the deal and its ultimate resolution. This allows us to use ex-post information to shape ex-ante gold targets, while still enforcing a strict temporal cutoff on what the model is allowed to see at prediction time. We next detail how we construct these gold targets for both deal reports and outcomes.

### 4.2.1 GOLD DEAL REPORTS

**Post-mortem analysis.** For each deal we run a dedicated post-mortem agent with access to full information up to and *after the resolution date* (subject to the overall training cutoff). The post-mortem agent is the only component in our system with access to open-web content, since information leakage is irrelevant once the deal has resolved. It produces three structured artifacts:

1. **General post-mortem**: a narrative of how the deal unfolded and which ex-ante signals (e.g., regulatory posture, shareholder alignment, financing quality) ultimately proved predictive.

2. **Timeline post-mortem**: an analysis of the realized sequence of events (filings, regulatory actions, litigation, shareholder votes, competing bids), highlighting which milestones accelerated, delayed, or derailed the deal.

3. **Market view post-mortem**: an interpretation of how the market priced the deal over time, including episodes where prices appeared to under- or overreact to new information.

**Hindsight-guided deal reports.** To construct a gold deal report for a specific forecast date $t$, we combine the (pre-$t$) forecasting-module context with the post-mortem artifacts and prompt a teacher model (GPT-5.1) to write a hindsight-guided deal report. We instruct it to: (i) reason *as if* making a forecast at time $t$, (ii) base every claim strictly on information available up to $t$ (with citations), and (iii) use the post-mortem only to identify, in hindsight, which pieces of information *should* have been considered salient at the forecast date, not to introduce future events or facts.

This procedure is inspired by process supervision (Uesato et al., 2022; Lightman et al., 2024; Luo et al., 2024): the post-mortem plays a similar role to a process reward model, guiding towards reasoning chains that are not only plausible but *causally* informative given the realized path. Importantly, even if some future information did leak into the gold

deal reports, it would not improve the model's quantitative performance because all evaluations are out-of-sample.

### 4.2.2 GOLD OUTCOME SUPERVISION

During training, the realized outcome $Y \in \{\textbf{Succeed}^+, \textbf{Fail}^+, \textbf{Fail}^-\}$ is known. However, training directly on one-hot labels encourages overconfident, brittle forecasts, especially when the deal is far from resolution. Instead, we construct smoothed three-way gold targets $p_t^*(\cdot)$ that blend hard labels with market-implied probability $p_t^m$.

Recall that $p_t^m$ is defined for the $+/-$ task and does not distinguish between the two positive outcomes $\textbf{Succeed}^+$ and $\textbf{Fail}^+$. We therefore build $p_t^*(\cdot)$ in two steps.

**Negative outcome.** We first define the gold probability of a negative outcome ($\textbf{Fail}^-$) as

$$p_t^*(\textbf{Fail}^-) = \alpha_t\big(1 - p_{t+7}^m\big) + (1 - \alpha_t)\,\mathbf{1}\{Y = \textbf{Fail}^-\}$$

where $\alpha_t \in [0, 1]$ is a time-varying smoothing parameter and $p_{t+7}^m$ is the 7-day *forward-shifted* market-implied probability of a $+$ outcome.[1] We set $\alpha_t$ to 0.3 at announcement and decay it smoothly to zero, so that targets rely more on market priors when uncertainty is high and more on the realized outcome as the deal approaches resolution.

**Positive outcomes.** The remaining probability mass, $1 - p_t^*(\textbf{Fail}^-)$, must be split between the two positive outcomes $\textbf{Succeed}^+$ and $\textbf{Fail}^+$. The market-implied probability $p_t^m$ only defines their sum, not the allocation. To determine this split, we use the teacher model (GPT-5) with access to post-mortem artifacts (Sec. 4.2.1), which produces a relative distribution over $\textbf{Succeed}^+$ and $\textbf{Fail}^+$.

**Days to completion.** We use actual days to completion as the gold target.

**Implementation Details** As the forecast LLM we select GPT-4o, finetuned via the OpenAI API. To help balance the dataset, we oversample terminated deals with weights determined on the validation set.

## 5 Results

**Baselines.** We compare our finetuned system to a variety of frontier models with access to identical context, plus baselines using market and deal-level features:

- **Market-implied probability.** $p_t^m$ from Sec. 3.2.1.
- **Calibrated market-implied probability.** Platt-scaled $p_t^m$ using train+validation data.
- **XGBoost on structured features.** Using $p_t^m$ and 26

---

[1]We use a short forward shift so that $p_{t+7}^m$ reflects how the market has digested information around date $t$, rather than the often noisy trading immediately after announcement.

*Table 3.* Overall results on $+/-$ outcome prediction. In all tables, superscripts indicate two-sided paired bootstrap significance of the difference relative to OURS (GPT-4O-FT) ($^\ddagger p < 0.01$, $^\dagger p < 0.05$, $^\circ p < 0.10$).

| # | Model | $\text{Brier}_{\mathbf{B}} \downarrow$ | $\text{Cal}_B \downarrow$ | $\text{Disc}_B \downarrow$ | $\text{Brier}_{\mathbf{S}} \downarrow$ | $\text{Brier}_{\$} \downarrow$ | MAPE $\downarrow$ | $(\hat{p}_t - p_t^m)^2$ | $\rho(\hat{p}_t, p_t^m)$ |
|---|---|---|---|---|---|---|---|---|---|
| 1 | Market $p^m$ | $0.229^\ddagger$ | 0.046 | 0.183 | $0.454^\ddagger$ | $0.319^\ddagger$ | – | – | – |
| 2 | Market (Platt) | $0.199^\ddagger$ | **0.016** | 0.183 | $0.301^\ddagger$ | $0.240^\ddagger$ | – | 0.063 | 1.000 |
| 3 | XGBoost | $\underline{0.186}^\dagger$ | $\underline{0.035}$ | 0.153 | $0.280^\ddagger$ | $0.246^\ddagger$ | – | 0.064 | 0.753 |
| 4 | Claude-37-Sonnet | $0.259^\ddagger$ | 0.146 | 0.108 | $0.276^\ddagger$ | $0.267^\ddagger$ | $0.433^\circ$ | 0.156 | 0.291 |
| 5 | Gemini-3-Flash | $0.250^\ddagger$ | 0.144 | **0.101** | $0.289^\ddagger$ | $0.282^\ddagger$ | $0.423^\circ$ | 0.158 | 0.319 |
| 6 | Gemini-3-Pro | $0.231^\ddagger$ | 0.118 | 0.110 | $0.276^\ddagger$ | $0.268^\ddagger$ | $0.497^\dagger$ | 0.149 | 0.339 |
| 7 | GPT-5.1 | $0.214^\ddagger$ | 0.110 | $\underline{0.103}$ | $0.244^\ddagger$ | $0.245^\ddagger$ | $0.465^\ddagger$ | 0.140 | 0.355 |
| 8 | GPT-4o | $0.201^\ddagger$ | 0.089 | 0.111 | $\underline{0.226}^\dagger$ | $\underline{0.234}^\ddagger$ | $0.463^\ddagger$ | 0.133 | 0.299 |
| 9 | GPT-4o + Isotonic | $\underline{0.186}^\ddagger$ | 0.075 | 0.110 | $\underline{0.219}^\ddagger$ | $\underline{0.228}^\ddagger$ | $0.463^\dagger$ | 0.149 | 0.314 |
| 10 | Ours (GPT-4o-FT) | **0.151** | 0.039 | 0.112 | **0.187** | **0.178** | **0.385** | 0.139 | 0.358 |

*Table 4.* Overall results on **Succeed/Fail** prediction.

| # | Model | $\text{Brier}_{\mathbf{B}} \downarrow$ | $\text{Cal}_B \downarrow$ | $\text{Disc}_B \downarrow$ |
|---|---|---|---|---|
| 1 | XGBoost | $\underline{0.164}^\ddagger$ | **0.013** | 0.151 |
| 2 | Claude-37-Sonnet | $0.239^\ddagger$ | 0.135 | **0.099** |
| 3 | Gemini-3-Flash | $0.227^\ddagger$ | 0.118 | 0.107 |
| 4 | Gemini-3-Pro | $0.210^\ddagger$ | 0.101 | 0.109 |
| 5 | GPT-5.1 | $0.173^\ddagger$ | 0.065 | $\underline{0.106}$ |
| 6 | GPT-4o | $0.177^\ddagger$ | 0.057 | 0.118 |
| 7 | Ours (GPT-4o-FT) | **0.126** | 0.019 | 0.107 |

*Table 5.* Qualitative example. Each scenario modifies the baseline context; $\Delta\hat{p}_t$ is averaged over 5 predictions.

| Scenario | Expected | $\hat{p}_t$ | $\Delta\hat{p}_t$ |
|---|---|---|---|
| Baseline (no event) | – | 82% | – |
| ISS recommends FOR | ↑ | 88% | +5% |
| ISS recommends AGAINST | ↓ | 73% | −9% |
| Major shareholder support | ↑ | 87% | +4% |
| Major shareholder opposition | ↓ | 54% | −28% |
| Bidder raises offer 10% | ↑ | 91% | +9% |
| Rumors of financing concerns | ↓ | 81% | −1% |

deal-specific features, including arbitrage profit, deal value, ownership structure, consideration mix, and competitive dynamics (Appendix A.2).

**Overall $+/-$ outcome prediction.** As reported in Tab. 3, our finetuned system (GPT-4o-FT) achieves the lowest Brier scores across all three weighting schemes, outperforming second-place models XGBoost on class-balanced Brier score ($\text{Brier}_B$, 0.151 vs 0.186), and second-place model GPT-4o on surprise and P&L weighted Brier scores ($\text{Brier}_{\mathbf{S}}$, 0.187 vs. 0.226; $\text{Brier}_{\$}$, 0.178 vs. 0.234). The surprise and P&L weighting schemes expose XGBoost's reliance on the market-implied probability feature–when the market is surprised, the XGBoost model performs poorly. In contrast to XGBoost, our system has low correlation with market implied probability (0.358 vs. 0.753).

**Finetuning substantially improves performance.** Even with access to curated context from our research agents, frontier models including Gemini-3-Pro, GPT-5.1, and GPT-4o underperform XGBoost on $\text{Brier}_B$ (Tab. 3, rows 6–8 vs. 3). Murphy decomposition suggests that these models achieve stronger *discrimination* (**Disc**) than XGBoost—they better separate higher- and lower-risk deals—but have weaker *calibration* (**Cal**). Finetuning substantially improves performance, especially calibration (row 10 vs. 8), resulting in the best overall Brier score. We observe similar patterns for **Succeed/Fail** prediction (Tab. 4), where our system again achieves the lowest Brier scores by a large margin.

**Finetuning outperforms post-hoc calibration.** The improvement from finetuning does not arise from post-hoc calibration alone: isotonic regression (Niculescu-Mizil & Caruana, 2005) improves the class-balanced Brier score of GPT-4o from 0.201 to 0.186 (row 9 vs. 8), but still falls well short of GPT-4o-FT at 0.151. Platt and temperature scaling similarly fail to close the gap (see Appendix B.1). While Murphy discrimination is broadly unchanged after finetuning, we note that it is a relatively coarse diagnostic (computed with 10 forecast bins), and may be missing finer-grained improvements in predictive quality.

**Finetuning improves days-to-close prediction.** Our system achieves a median absolute percentage error (MAPE) of 0.385 on completed deals, compared to 0.463 for GPT-4o.

**Qualitative example.** To illustrate how the model adjusts probabilities in the response to information, in Tab. 5 we intervene on an example deal, modifying the deal context by hand. Across six example scenarios spanning three causal mechanisms–proxy advisory influence (ISS recommendation), shareholder voting, and deal terms–in each case the model adjusts $\hat{p}_t$ in the expected direction. The largest effect (shareholder opposition) reflects a realistic constraint.

## 5.1 Ablation studies

In Tab. 6 we report ablations to evaluate how much each component supports the system.

**Scaling the training set is likely to yield further gains.** Training on only 50% of the data (row 2) – stratified by region to maintain diversity – degrades $\mathbf{Brier}_B$ from 0.151 to 0.177, suggesting that performance has not yet saturated with respect to training set size.

**Dataset balancing requires careful tuning.** During training we oversampling terminated deals, multiplying the frequency of terminated negatives and positives by 1.65 and 1.25, respectively. Removing oversampling during finetuning (row 3) degrades $\mathbf{Brier}_B$ from 0.151 to 0.170. Oversampling too aggressively, with weights (2.0, 1.65) degrades $\mathbf{Brier}_B$ to 0.163 (row 4).

**Hindsight guidance and market smoothing provide gains.** We also isolate the contribution of our supervision recipe. Removing hindsight-guided postmortem supervision (row 5) degrades $\mathbf{Brier}_B$ from 0.151 to 0.183, while removing market-probability smoothing (row 6) degrades it to 0.181.

**Specialized research agents add value.** Ablating the information provided by specialized research agents (Deal Card only, row 7) degrades $\mathbf{Brier}_B$ from 0.151 to 0.172. In rows 8–14 we add back one research agent at a time (corresponding to Tab. 2), in each case observing improvements over Deal Card only (row 7) that indicate that each agent contributes useful information. Since research agent responsibilities are informed by merger-arb specialists, we do not enforce that agents retrieve non-overlapping information; nevertheless we find that removing agents individually (Appendix B.2) generally worsens performance, although with limited statistical significance.

## 5.2 Deal reports

We deploy this system primarily as a decision-support tool for discretionary investors, making the reasoning and grounding of the reports especially important. In this section, we study the quality of the generated deal reports.

**Grounding.** Using an LLM-based grader (Appendix D.4), we assess the factual grounding of generated deal reports by: (1) identifying claims in the report, and (2) assessing the level of evidence for each claim. Our finetuned (GPT-4o-FT) model averages 33 claims per report with an unsupported-claim rate of 0.1%, compared with 9 claims and 0.3% for the GPT-4o base model, suggesting that finetuning with hindsight-guided supervision does not introduce hallucinations or reduce factual grounding.

**Rubric-based assessment.** A key application of our system is forming a Day-1 view in response to a deal announce-

*Table 6.* Impact of system ablations on $+/-$ outcome prediction.

| # | Model | $\mathbf{Brier_B} \downarrow$ | $\mathrm{Cal}_B \downarrow$ | $\mathrm{Disc}_B \downarrow$ |
|---|---|---|---|---|
| 1 | Ours (GPT-4o-FT) | **0.151** | 0.039 | **0.112** |
| 2 | …50% data | $0.177^{\ddagger}$ | 0.062 | 0.115 |
| 3 | …No oversampling | $0.170^{\ddagger}$ | 0.058 | **0.112** |
| 4 | …More oversampling | 0.163 | 0.042 | 0.121 |
| 5 | …No hindsight guide | $0.183^{\ddagger}$ | 0.062 | 0.121 |
| 6 | …No $p_t^m$ smoothing | $0.181^{\ddagger}$ | 0.081 | 0.100 |
| 7 | …Deal Card only | $0.172^{\dagger}$ | **0.036** | 0.137 |
| 8 | …Filings Analysis | 0.163 | 0.040 | 0.123 |
| 9 | …Stakeholder Own. | $0.158^{\circ}$ | 0.042 | 0.117 |
| 10 | …Market View | $0.165^{\dagger}$ | 0.043 | 0.122 |
| 11 | …Current Climate | $0.170^{\circ}$ | 0.040 | 0.130 |
| 12 | …Precedent Mergers | $0.175^{\dagger}$ | **0.036** | 0.140 |
| 13 | …Regulatory Risk | 0.167 | 0.039 | 0.128 |

ment. To evaluate whether our reports capture the resulting decision-relevant issues within hours of announcement, we construct deal-specific checklists of the key ex-ante risks and mitigants visible at the initial forecast date and score whether each report identifies them (rubric construction details and examples in (rubric construction and grading details in Appendices D.1 and D.2). GPT-4o-FT achieves 50.2% rubric coverage vs. 21.5% for the base model ($+28.7$ pp, $p<0.001^{***}$). Coverage remains partial even for the finetuned model, reflecting the difficulty of identifying non-obvious, analyst-level, and idiosyncratic insights from Day-1 context.

**Failure analysis.** Appendix D.3 analyzes the largest GPT-4o-FT forecasting errors. Most arise from missing forecast-date context rather than faulty reasoning. Web search would reduce this gap, but we disable it to avoid information leakage (Section 3.3).

## 6 Conclusion

We present an LLM-based merger-arbitrage forecasting system that combines specialist research agents with hindsight-guided finetuning. It outperforms frontier models, market-implied probabilities, and XGBoost on held-out deals; ablations trace the gains to specialist context, hindsight-guided supervision, target smoothing, class balancing, and training-set scale.

## Impact Statement

This paper studies language-model-based forecasting for announced public M&A transactions. The main positive impact is improved decision support: the system may help analysts process complex public information more efficiently and consistently. Potential risks include over-reliance on model outputs, increased informational asymmetry, and misuse in trading contexts. Our system is intended for human decision support, not autonomous trading, and should be used with appropriate oversight.

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

# A Hyperparameters

## A.1 Forecasting LLM

We fine-tuned a supervised forecasting model using the OpenAI supervised fine-tuning API on `gpt-4o-2024-08-06`.

Table 7 contains fine-tuning and inference settings.

| Item | Value |
|------|-------|
| Fine-tuning method | OpenAI supervised fine-tuning API |
| Base model | `gpt-4o-2024-08-06` |
| Teacher model for gold outputs | `gpt-5.1` |
| System prompt | "You are a superforecaster with merger arbitrage expertise." |
| Epochs | 1 |
| Batch size | 2 |
| Learning rate multiplier | 1 |
| Inference temperature | 0.2 |
| Inference top-$p$ | 1.0 |
| Inference max tokens | 16384 |

*Table 7.* Fine-tuning and inference settings for the GPT-4o supervised finetuned model.

## A.2 XGBoost Baseline

Our strongest structured baseline is a three-class XGBoost classifier trained on point-in-time structured deal features.

To reduce class imbalance, inverse-frequency class weights are applied at training time and capped at a maximum ratio of $10\times$ between the rarest and most common class weights.

The feature set combines market-implied information with point-in-time deal-structure fields, as listed in Table 8. The XGBoost hyperparameters are included in Table 9.

| Feature group | Variables |
|---------------|-----------|
| Market-implied features | market-implied probability of success, normalized merger spread, normalized downside-to-break risk |
| Economics / valuation | announcement premium, current premium, log transaction value, arbitrage profit estimate |
| Ownership / control | fraction of target shares sought, target shareholder ownership in the post-merger entity |
| Consideration mix | cash component present, stock component present, contingent consideration present, cash share of total consideration |
| Legal / process terms | financing condition present, MAE clause present, dissenters' rights present, go-shop period length, drop-dead date present |
| Bid dynamics | friendly bid indicator, hostile bid indicator, unsolicited bid indicator |
| Deal type indicators | acquisition indicator, merger indicator, divestiture indicator |
| Termination fees | acquirer-to-target termination fee / deal value, target-to-acquirer termination fee / deal value |

*Table 8.* Features used by the `XGBoost` baseline.

# B Additional Results

## B.1 Additional post-hoc calibration baselines

We include additional calibration baselines in Table 10.

## B.2 Selected leave-one-out ablations

We include leave-one-out ablations in Table 11.

| Hyperparameter | Value |
|---|---|
| Objective | `multi:softprob` |
| Number of classes | 3 |
| Max depth | 5 |
| Min child weight | 3 |
| Gamma | 0.1 |
| Learning rate | 0.05 |
| Number of trees | 50 |
| Subsample | 0.8 |
| Column subsample | 0.8 |
| Optimization metric | `mlogloss` |
| Missing-value handling | Training-median imputation for numeric features |
| Class weighting | Inverse-frequency weights |

*Table 9.* Hyperparameters for XGBoost, tuned on the validation set.

*Table 10.* Additional post-hoc calibration baselines for GPT-4o on $+/-$ outcome prediction.

| Model | $\text{Brier}_\mathbf{B} \downarrow$ | $\text{Cal}_B \downarrow$ | $\text{Disc}_B \downarrow$ | $\text{Brier}_\mathbf{S} \downarrow$ | $\text{Brier}_\mathbf{\$} \downarrow$ | $\textbf{MAPE} \downarrow$ | $(\hat{p}_t - p_t^m)^2$ | $\rho(\hat{p}_t, p_t^m)$ |
|---|---|---|---|---|---|---|---|---|
| GPT-4o + Platt | $0.214^{\ddagger}$ | 0.101 | 0.111 | $0.247^{\ddagger}$ | $0.256^{\ddagger}$ | $0.463^{\dagger}$ | 0.148 | 0.302 |
| GPT-4o + Temp-Scale | $0.222^{\ddagger}$ | 0.109 | 0.111 | $0.254^{\ddagger}$ | $0.263^{\ddagger}$ | $0.463^{\dagger}$ | 0.149 | 0.300 |

*Table 11.* Selected leave-one-out ablations from the full GPT-4o-FT system on $+/-$ outcome prediction.

| Model | $\text{Brier}_\mathbf{B} \downarrow$ | $\text{Cal}_B \downarrow$ | $\text{Disc}_B \downarrow$ | $\text{Brier}_\mathbf{S} \downarrow$ | $\text{Brier}_\mathbf{\$} \downarrow$ | $\textbf{MAPE} \downarrow$ | $(\hat{p}_t - p_t^m)^2$ | $\rho(\hat{p}_t, p_t^m)$ |
|---|---|---|---|---|---|---|---|---|
| FT w/o Financing Risk | 0.155 | 0.037 | 0.118 | 0.206 | 0.194 | 0.405 | 0.137 | 0.373 |
| FT w/o Filings Analysis | 0.156 | 0.036 | 0.120 | 0.190 | 0.193 | 0.404 | 0.136 | 0.366 |
| FT w/o Stakeholder Own. | 0.158 | 0.043 | 0.114 | 0.201 | 0.188 | 0.393 | 0.138 | 0.352 |
| FT w/o Deal Card | 0.165 | 0.048 | 0.117 | 0.202 | 0.195 | 0.397 | 0.140 | 0.354 |
| FT w/o Current Climate | 0.156 | 0.038 | 0.120 | 0.201 | 0.188 | 0.412 | 0.136 | 0.369 |
| FT w/o Precedent Mergers | 0.161 | 0.043 | 0.117 | 0.208 | 0.195 | 0.400 | 0.138 | 0.361 |
| FT w/o Market View | 0.154 | 0.039 | 0.115 | 0.204 | 0.194 | 0.396 | 0.138 | 0.361 |
| FT w/o Gap Analysis | 0.164 | 0.043 | 0.120 | 0.213 | 0.202 | 0.413 | 0.141 | 0.336 |
| FT w/o Regulatory Risk | 0.157 | 0.036 | 0.120 | 0.207 | 0.190 | 0.415 | 0.136 | 0.371 |

# C  Prompt Templates

## C.1  Forecasting LLM Prompt

---

**Forecasting LLM Prompt**

You are a superforecaster working on a merger arbitrage team at a hedge fund. You have an excellent track record of predicting deal close probabilities due to your comprehensive, detail-oriented, leave-no-stone-unturned research style. You also have access to research collected from a state-of-the-art research library with millions of financial documents.

You are asked to create a forecast for the following deal: {`merger_acquisition_desc`}.

You are generating a report as of {`today_date`}. Your job is to produce a forecast as if you were working on that date, using **only** information available up to that date. It is absolutely critical that you do not use any information from after {`today_date`} in your analysis or forecast.

**Background on Superforecasting**

Superforecasting is the disciplined art of: breaking questions into component parts; establishing base-rate priors; assigning explicit probability estimates; relentlessly incorporating new evidence via Bayesian updating; actively seeking disconfirming facts; pooling diverse perspectives; tracking forecast accuracy; and maintaining intellectual humility while remaining doggedly self-critical.

**Approach**

You are a merger arbitrage expert, and so is the reader of your report. Use this shared expertise: think through the lens of a merger arb analyst and employ the appropriate technical language. This is advanced research, not intended for a general audience.

Carefully examine all information already available to you, paying close attention to dates, timelines, the deal mechanics, the regulatory environment, and relevant precedents and base rates. Conduct a thorough and rigorous analysis covering potential red and green flags, regulatory hurdles, bidding wars, financing issues, and any other factors that could affect the deal close probability.

You will be given information already collected about the deal, including precedent mergers, stakeholder ownership, risk analysis, market views, filings analysis, current climate, and any other relevant information.

**Do not simply defer to the companies' guidance or expert consensus.** Analyze all pieces of information and facts critically. Pay close attention to the details. Think step by step.

**Forecast Output Requirements**

You must produce the following:

1. **Probability of a positive outcome for the target company**: `<probability>...</probability>`
2. **Probability of the deal closing**: `<close_probability></close_probability>`
3. **Estimated number of days until close**: `<days_until_close>...</days_until_close>`
4. **Detailed Report**: A thorough written analysis explaining your reasoning, the red and green flags identified, key details, probability breakdown, and how you arrived at your final forecast.

**1. Probability of Positive Outcome for the Target Company**

Provide the probability (0–1) that the deal will successfully complete (i.e., the merger/acquisition closes) **or** that the target company receives a better/competing offer.

- **Required:** Include the single XML tag `<probability>p</probability>`, where `p \in [0,1]` represents your primary headline probability. This tag must appear **exactly once** in your answer.
- **Additionally**, provide the standalone close probability as `<close_probability>q</close_probability>`.

**Note:** If {`merger_acquisition_desc`} is currently only rumored or under strategic review, interpret the probability as the likelihood of (a) a formal announcement **and** successful closing, or (b) no announcement of this specific deal but the target achieving a positive outcome through another offer.

Clearly articulate the breakdown of your probability calculation and the scenarios considered.

**2. Estimated Number of Days Until Close**

If the deal is predicted to close, provide a point estimate of the number of **calendar days from {`today_date`}** until closing (or counterfactual closing if you predict termination) within `<days_until_close>...</days_until_close>` tags. In the detailed report, also provide a range.

**3. Detailed Report**

---

Provide a detailed written report explaining your full reasoning and analysis. The report should include:

- **Full thought process**: Walk through your analysis step by step.
- **Evidence weighting**: Discuss the evidence considered and how you weighed different factors.
- **Base rates and precedents**: Analyze relevant base rates and precedent deals (if available) to establish a prior, then update based on deal-specific information.
- **Red and green flags**: Clearly identify and explain each flag.
- **Scenario analysis**: Present distinct possible outcomes with associated probabilities.
- **Risks and mitigations**: Cover the details a merger arb analyst would care about.
- **Probability breakdown**: Explain precisely how you arrived at each output.
- **Citations**: All facts used must be traceable to the input context via citations such as `[market-2]`, `[precedent-5]`, etc. You may use the phrase *"From my internal knowledge, I believe..."* only for generic, atemporal common-sense priors and rough estimates (e.g., typical regulatory processing windows) — **never** to retrieve time-dependent facts (e.g., who is currently in office) or any characteristics, data points, or outcomes specific to this deal beyond what is provided in the input.

**Use Markdown formatting throughout, including bullet points, numbered lists, and section headers to organize your report clearly.**

**Input Context**

Here is the information already collected about the deal:

**Existing Context:** `{existing_context}`

## C.2 Data Curation Prompts

---

**Post-Mortem Analysis**

We are curating a post-mortem analysis of a merger and acquisition (M&A) deal.

Using your expertise in merger arbitrage and the tools at your disposal, thoroughly analyze the events surrounding the deal.

Your post-mortem analysis will be used by a teacher model to train a forecasting model to improve its future predictions, so your research should be thorough and detailed. Look for nuanced signs.

The predictor model is asked to predict the following about the deal:

1. **Deal Probability:** The likelihood of the deal being (1) successfully completed, (2) terminated with no better offer, or (3) terminated with a better offer (for the target).
2. **Expected Days to Completion:** The expected number of days from the announcement date until the deal is completed (if predicted to complete).
3. **Key Risks and Mitigations:** The key risk factors that could derail the deal and the potential mitigations to address those risks.

In your analysis, research the following to help the teacher create a forecast that is guided by hindsight but also grounded in the information available at the time:

- What telltale signs were there in the deal terms, company backgrounds, market conditions, or regulatory environment that could have indicated the eventual outcome of the deal?
- Were there any red flags or positive indicators that the market overlooked or misinterpreted?
- How did the actions and communications of the involved companies, regulators, or other stakeholders influence or reveal the deal's trajectory?
- What factors (e.g., precedents, market overlap, etc.) could have been analyzed to correctly predict the key gating item, potential termination reason, and/or days to completion?
- What terms of the merger agreement were particularly favorable or unfavorable, and how did they impact the deal's success or failure?
- For deals that faced antitrust scrutiny, what specific arguments did regulators use, and how could these have been anticipated?
- How did these signs change over the timeline of the deal, from rumor to announcement to resolution?

You should:

- Identify the risks/mitigations that actually arose (or almost arose) during the timeline of the deal. Try to perform an independent analysis. For example, if the deal received a DOJ second request, identify the reason cited and then work backwards to find the signs that could have predicted this. If the deal was terminated due to financing issues, identify the reasons and then work backwards to find signs that could have predicted this.
- Identify the potential risks/mitigations that were mentioned in research or media in the days or weeks following the deal announcement. We are looking for key observations, details, or insights that were mentioned or discussed. These will likely be buried in the text; carefully read through it to find them. The risks/mitigations should especially include non-obvious, potentially buried, insights.
- Keep each phase of the deal in mind (rumor, announcement, regulatory review, shareholder approval, . . . , closing) and highlight which signs were relevant at which phase. If a sign was only known after a certain point in time, make that clear.
- Focus on facts and insights, not opinions. Avoid including opinions or subjective views unless they are directly tied to a factual insight or event that could have been known at the time.

And produce:

1. An overall post-mortem analysis of the deal (3–5 concise sentences).
2. A post-mortem of the deal's timeline of events. The goal is to explain how the deal ended up taking the amount of time it did to reach its outcome. Produce a concise timeline like this:
    - Announcement was made on YYYY-MM-DD.
    - Proxy was filed on YYYY-MM-DD. This typically takes X days after announcement. It was slower in this case because of REASON.
    - Vote was held on YYYY-MM-DD, as laid out in the proxy.
    - Holiday season caused delays from YYYY-MM-DD to YYYY-MM-DD.
    - Regulatory approval was received on YYYY-MM-DD after an extended review due to REASON.
    - Conclusion: took X days from announcement, with . . .

    This output should be a bullet-point list of the key events that determined the timeline of the deal, with brief explanations where relevant. Include signs that could have predicted the timeline outcome.
3. A post-mortem of the market's probability and view of the deal. For each of the dates below, find or compute the market-implied probability of deal completion (e.g., from merger arbitrage spreads or analyst-implied probabilities, if available). Provide a

qualitative interpretation of why the market viewed the deal that way on that date. `{dates}`

`<post_mortem_analysis>` Produce a post-mortem analysis of the deal. Begin with a one-sentence summary of the deal's trajectory and outcome. Then write 3–5 concise sentences focusing on the most critical insights. Do not provide any forecasting rules; focus on finding the pieces of the mosaic that would have helped predict the outcome of this specific deal. `</post_mortem_analysis>`

`<timeline_post_mortem>` Produce a post-mortem of the deal's timeline of events. `</timeline_post_mortem>`

`<market_post_mortem>` Produce a post-mortem of the market's probability and view of the deal. `</market_post_mortem>`

`<rubric>` Produce a concise rubric of up to 4–6 key insights, risks, or mitigations that the model should have identified during this deal (prior to the resolution). Each item in the rubric should be 1–2 sentences long. Especially look for items that only one or a few authors identified (but also include obvious ones if they are particularly important to this deal). The rubric should be formatted as a numbered list. Be concise and to the point. Each item in the rubric must be a fact that could have been known as of a certain date. For example, if a DOJ investigation happened months later, it cannot be included unless there were clear earlier signs (e.g., a prior statement of intent to reduce competition from an older earnings call). Try to clearly state the date or phase of the deal when the fact or insight was known or could have been known.

A rubric item cannot be an individuals view or opinion. It must be a fact or insight that would be helpful for merger arbitrage. Each item must be sourced from your own data analysis, research, or statements from the companies or regulators involved in the deal. `</rubric>`

**Deal Context:** `{deal_context}`

---

**Golden Forecast Generation**

You are a teacher model whose outputs will be used to fine-tune a student model. The student's task is to forecast the likely outcomes for a merger or acquisition as of the date implied by the input prompt.

**High-level objective**

Act like a teacher who privately has full hindsight (including post-mortem context, ground truth, and internal calibration signals) and uses it only to internally calibrate a high-quality forecast. However, the output you write must be a student-facing, step-by-step `<walkthrough>` and final report that a very strong analyst could plausibly have produced using only the **INPUT PROMPT** as of the specified date, with zero explicit or implicit leakage of any teacher-only context.

You will be given the same input prompt that will also be given to the student model. This input prompt contains research already done about the deal (for example, deal terms, filings analysis, precedent transactions, market views, regulatory risk, and related materials). Your final output may cite and use only the information provided in the input prompt. Do not use any external information or knowledge, because the purpose of this task is to teach the student model how to make forecasts using only the given information.

**Your job**

Produce a detailed forecast report for the deal that includes:

- A thorough scenario analysis of possible deal outcomes and their likelihoods.
- The outputs requested in the prompt: probability of a positive outcome, probability of closing, estimated days to close, and a detailed report with red flags and green flags.
- A report covering the details that a merger-arbitrage trader would want to know as of the given date, which are key risks and mitigants, including relevant rubric items *if and only if* they are supported by the information in the input prompt. If a rubric point cannot be known from the input prompt, do not include it in the report.

The input prompt may rely primarily on contemporaneous documents, while the rubric or other teacher-only context may reflect later knowledge. Only cite what appears in the input prompt.

**Teacher-only context you may receive**

The teacher-only context may include:

- A general post-mortem analysis: a concise narrative of the deal's trajectory and outcome, plus key signs that, in hindsight, could have helped predict that outcome.
- A timeline-specific post-mortem: a bullet-point breakdown of major events and their timing from announcement to outcome, with commentary on what actually drove the realized timeline.
- A market-specific post-mortem: a description of how the market viewed and priced the deal over time, sometimes including qualitative or approximate market-implied probabilities on a small set of reference dates based on merger-arbitrage spreads, commentary, and observed price action.
- A rubric: 4–6 concise items highlighting key risks, mitigants, or nuanced insights that a strong merger-arbitrage analyst should have identified before resolution, with emphasis on non-obvious, alpha-generating observations.

**Important note:** The post-mortem is generated only after the deal has completed or terminated. Therefore, not all information in the post-mortem would have been available as of the date specified in the input prompt. Use teacher-only context only for internal calibration. If a sign was available as of the input date and is also present in the input context, you may highlight it. If a sign emerged only later in the deal life cycle, do not include it in the student-facing output.

**Notes on the probability forecast**

- The student model (and therefore your report) must predict the probability of a positive outcome for the target company, where a positive outcome means either:
  - the deal closes successfully, or
  - the target receives a better or competing offer.
- You will be provided with the ground-truth outcome of the deal. This is for internal calibration only.
- In the walkthrough and report, do not simply return the ground-truth outcome or state your probability without analysis. You must arrive at your probability estimate through careful analysis of the input prompt only.
- Walk through your process for arriving at the probability estimate in detail.
- Use the implied probability anchor provided in the Additional Context as a quantitative internal starting point. Reconcile your final probability to this anchor unless the input facts justify a deviation. Do not cite or mention the anchor in the student-facing output.
- The implied probability anchor is constructed as follows:
  - On each trading day

$$t$$

after announcement, the target's share price $S_t$ is treated as lying between a downside path $D_t$ (the stand-alone break value) and an upside path $U_t$ (the value if the deal or a higher bid succeeds). The market-implied probability is defined as $p_t^m = \text{clamp}_{[0,1]}\left(\frac{S_t - D_t}{U_t - D_t}\right)$. Thus, $p_t^m$ measures how far the current price lies between the break price and the deal value.

– $D_t$ is constructed by anchoring to the 20-day pre-announcement average and allowing it to move with a broad equity index scaled by the target's pre-announcement beta.

– $U_t$ is constructed directly from the prevailing deal terms (cash, stock, or mixed), including discounting for expected time-to-close using risk-free rates and updating whenever terms are amended.

– To align with the student prompt, the system does not always read off $p_t^m$ exactly on the as-of date implied by the input. Instead, it:

  * takes the as-of date from the input prompt, then
  * looks up $p_t^m$ on a reference date equal to that as-of date plus a small configured offset of `move_forward` calendar days.

  This step allows the anchor to reflect how the market prices the deal shortly after the information in the input prompt is available and smooths short-term noise on the exact as-of date.

– The raw $p_t^m$ is informative but noisy, especially early in the deal lifecycle. Therefore, the implied probability anchor is defined as a calibrated blend: $\text{anchor}_t = \alpha(t)\, p_t^m + \left(1 - \alpha(t)\right) y$, where $y$ is the realized outcome label (1 for a positive outcome, 0 for a negative outcome), and $\alpha(t)$ is a weight that decays with time since announcement over a fixed horizon.

– For purposes of this task, you should think of the implied probability anchor as an internal estimate of the market probability of a positive outcome around the forecast date, calibrated toward the eventual outcome as the deal progresses over time.

- In the displayed student-facing analysis (both the `<walkthrough>` and the detailed report), do *not* use market spread or market-implied probability to derive the probability estimate. The goal is to arrive at the final probability and expected days to outcome using a fundamental, mosaic-style analysis of the deal characteristics presented in the input prompt.

- The goal is to produce a forecast that is grounded in the input, is well calibrated, and is more informative than market-implied probability would be at test time.

**Biases to avoid**

- Just because an event did not happen (for example, no competing bid emerged), that does not mean the probability of that event was zero at the time. Likewise, if an event did occur, that does not mean it was certain ex ante.

- Always ground your answer in the input prompt to avoid leakage and to avoid teaching the student model to hallucinate.

- If there is contradictory information between the input prompt and teacher-only context, follow the input prompt and base the student-facing analysis on that.

- If there are contradictions within the input prompt itself, explicitly highlight them and reason through them.

- Do not simply adopt the view of existing research, or risk verdicts included in the input prompt. Treat them as inputs, not conclusions. You must do your own independent analysis of the facts in the input prompt.

**Notes on the days to outcome estimate**

- You must produce an estimated number of days until close inside `<days_until_close>...</days_until_close>`, even if you predict that the deal will terminate.

- If you predict termination, then `days_until_close` should be a counterfactual estimate of how long the deal would likely have taken to close if it had proceeded to closing.

- Interpret "days until close" as the number of calendar days from `{current_date}` (the as-of date in the input prompt) until closing, or until the counterfactual closing date if the deal does not close.

- You will be provided with:
  1. the ground-truth number of days between announcement and outcome, and
  2. a summary of the actual timeline of events.

  Use these only as internal calibration aids. Do not reveal or quote them.

- Your estimate must be consistent with the facts in the input prompt.

- The walkthrough must show explicit steps and calculations.

To show your breakdown of the estimated days to close, do the following:

- You may draw inspiration from the detailed timeline of events provided in teacher-only context, but do not copy it or reveal it.

- For each event in your timing analysis, write full sentences:
  – If it is a fact, cite the relevant input tag(s).
  – If you are estimating a duration or likelihood not stated in the input, begin the sentence with ``From my internal knowledge, I believe ...'' and give an $X - Y$ day range tied to as-of-date facts.

- – The phrase ``From my internal knowledge, I believe ...'' may be used only for rough time estimates and truly atemporal common-sense background assumptions. It must not be used to supply any deal-specific fact, any circumstantial inference, any time-dependent fact, any market-condition assumption, any party-specific characteristic, any regulatory or political fact tied to the period, or any other information not explicitly contained in the input prompt.
  - – If information is missing, write ``What is unknown is ...'' and state the gap.
- Create three scenarios: Accelerated, Base, and Slow.
- Sum event durations for each scenario.
- State explicitly how you handle concurrency (for example, taking the slower of antitrust review versus proxy process, then adding mechanical closing steps).
- Assign scenario weights that sum to 1.0, with one-line justifications anchored to the input.
- Compute the weighted-average number of days and provide a point estimate inside `<days_until_close>`.

**Constraints for the teacher model**

- Use only facts from the **INPUT PROMPT** for analysis and citations.
- Cite using the existing `[module-X]` tags. Do not add URLs or new tags.
- If a module's content is missing a tag, you may cite the entire module by using the module name as a tag (for example, `[deal_card]`).
- For each tag that the input prompt requires:
  - – Include it **exactly once** in your entire output.
  - – Do *not* include any of the prediction tags `<probability>`, `<close_probability>`, or `<days_until_close>` inside the `<walkthrough>` block.
  - – **Critical output order:** output the prediction tags **immediately after** `</walkthrough>` and **before** the detailed report. This ensures predictions are captured even if the output is truncated.
  - – The content of `<probability>` must be a single bare number in $[0, 1]$ with no extra text (for example, `0.63`, not `63%` or `0.63 (63%)`).
  - – If `<close_probability>` is requested, its content must also be a single bare number in $[0, 1]$ with no extra text.
  - – The content of `<days_until_close>` must be a single numeric value (integer or decimal) with no units or commentary (for example, `145`, not `145 days` or `about 145`).
- You may restate or interpret these values elsewhere in words (for example, "around 60–65%"), but the tag contents themselves must remain pure numbers.

**Required output format**

Follow this structure exactly:

```
<walkthrough>
[Your step-by-step reasoning and analysis here...]
</walkthrough>

<probability>X.XX</probability>
<close_probability>X.XX</close_probability>
<days_until_close>XXX</days_until_close>
```

**Detailed Report**

```
[Your full merger-arbitrage analysis report here...]
```

**Citation protocol**

These rules apply to both the `<walkthrough>` and the detailed report:

- Support each non-trivial factual claim with one or more citations using the exact bracketed source tags embedded in the input prompt (for example, `[market-2]`, `[reg-3]`, `[precedent-5]`).
- Place citations at the end of the sentence or bullet they support.
- Never place citations inside `<probability>`, `<close_probability>`, or `<days_until_close>`.

**Forbidden phrases and leakage guardrails for student-facing text**

These rules apply to all student-facing text, including the `<walkthrough>`:

- Do not use or mention phrases or concepts that would leak teacher-only context or hinder learning, including:
  - – post-mortem

     – `implied anchor`
     – `market-implied probability`
     – `ground truth outcome`
     – `hindsight`
     – `teacher-only context`

- Do not reference specific future events, rulings, or dates that would only be known after {`current_date`}, even if they appear in teacher-only context.
- If you want to highlight a risk that appears only in the rubric or post-mortem, do so only indirectly and only if it is plausibly motivated by the as-of-date facts in the input prompt. For example, you may say, ``I would look for additional information on X'', but only if $X$ is reasonably suggested by the input prompt.
- The goal is to sound like an unusually strong as-of-date analyst, not like someone using future information.

**Walkthrough versus internal chain-of-thought**

- You may use teacher-only context in your *internal* reasoning, but you must never reveal or allude to it in any student-facing text.
- Do not reveal internal chain-of-thought or inner monologue.
- Instead, produce a student-facing `<walkthrough>` that breaks the problem into explicit steps, checks, confirmations, and calculations (for example, "Step 1:", "Check:", "Therefore:").
- The `<walkthrough>` should show how a careful analyst could, in principle, arrive at the forecast using only the **INPUT PROMPT**.
- Treat the walkthrough as structured scratch work based solely on the provided context, not as a polished final essay.
- It is acceptable, and encouraged, for the walkthrough to include course corrections or brainstorming, such as noticing inconsistencies across tagged sections and updating the view accordingly.
- However, every step in the walkthrough must be explicitly grounded in facts from the input prompt, with citations, and must not rely on or mention any teacher-only context.
- The walkthrough is intended to teach the student model a process, so do not simply copy the detailed report into the walkthrough.

**Base-rate requirement**

If the input prompt includes precedents or base-rate data, include a brief base-rate priors subsection in both the `<walkthrough>` and the detailed report that:

1. attempts to estimate base rates to the extent possible and useful from the provided data,
2. explains why some precedents may not be good comparables or how the current deal differs, and
3. states whether the current deal should sit above, below, or in line with those priors.

If no useful precedents are provided, explicitly say so and skip this step.

You may use the phrase ``From my internal knowledge, I believe ...'' only for generic, atemporal common-sense priors and rough estimates (for example, typical regulatory processing windows), and never to retrieve time-dependent facts or any specific characteristics, data points, or outcomes about this deal beyond what appears in the input prompt.

**Today's date**

Current Date = ``{`current_date`}''

Phrase your report as of this date. Do not include any information that would not have been known as of this date. Use present tense when referring to this date.

**Input Prompt (Given to Student Model — Frame Your Response as if Replying to This)**

{`input_prompt`}

**End of Input Prompt**

**Additional Context (for Teacher Model Only; Do Not Include in Student Output)**

- Days to Outcome Ground Truth (days between {`current_date`} and outcome or termination): {`days_to_outcome_ground_truth`}
- Market-Implied Probability $p_t^m$ as of {`current_date`} (plus {`move_forward_days`} days if configured): {`implied_probability_raw`}
- Calibrated Implied Probability Anchor $\alpha(t)p_t^m + (1 - \alpha(t))y$ as of {`current_date`}: {`implied_probability_anchor`}
- More Detailed Timeline of Events from Announcement to Outcome: {`detailed_timeline_of_events`}

- Ground Truth Outcome: {`ground_truth_outcome`}
    (`completed`/`terminated_positive`/`terminated_negative`)
    `completed` means the deal closed successfully; `terminated_positive` means the deal terminated but the target received a higher bid; `terminated_negative` means the deal terminated without a higher bid.
- Post-Mortem Analysis: {`post_mortem_analysis`}
- Key Risks/Mitigants Rubric: {`rubric`}

Provide the forecast report. Do not include any other commentary. Simply provide the report.

# D   Qualitative Evaluation

This appendix details the rubric construction and grading procedure used in the rubric-based assessment of Section 5.2, followed by an error analysis of the worst forecasting errors made by GPT-4o-FT on the test set.

## D.1   Day-1 Rubric Curation

For each deal in the test set, we generate a Day-1 rubric by prompting `gpt-5` with the deal-announcement context and access to a set of research tools. The model is asked to enumerate non-obvious risks, mitigants, or deal-specific facts that were knowable on or shortly after the announcement date; research published within roughly one week of announcement is admissible, but facts that only became public later are excluded unless there were documented signals at announcement. The full prompt is shown below. We deliberately separate this prompt from the in-training post-mortem rubric block (see "Post-Mortem Analysis" in Appendix C.2), which is used during data curation; the prompt below is restricted to Day-1 context.

---

**Day-1 Rubric Generator (`gpt-5`)**

We are building an evaluation rubric for a merger-arbitrage forecasting model.

The model is given a Day-1 context (deal announcement + publicly available information as of the announcement date) and must forecast:

1. Deal completion probability (completed / terminated / terminated with higher bid).
2. Expected days to close.
3. Key risks and mitigations.

Your task is to identify non-obvious facts and insights that a sophisticated merger-arb analyst should have caught from information available within the first ~week after the announcement date (`{announce_date}`).

**Temporal constraint (CRITICAL):**

- Each rubric item must be a fact that could have been known on or shortly after the announcement date (`{announce_date}`).
- Do NOT include facts that only emerged months later (e.g., a DOJ second request filed 6 months later) UNLESS there were clear documented signals at announcement (e.g., prior public statements by the acquirer about wanting to reduce competition).
- If an item is only known after a certain date, state that date clearly.

**What to look for (focus on non-obvious catches):**

- Important, unusual, or onerous merger-agreement terms (e.g., termination-fee structure, matching rights, fiduciary-out triggers, go-shop provisions, regulatory covenants).
- Regulatory red or green flags visible at announcement.
- Financing risk signals: debt-package details, bridge-loan terms, market conditions, acquirer leverage.
- Governance / shareholder-structure issues or benefits: controlling shareholders, voting agreements, conflicted directors, required vote thresholds, friendly relationships with key stakeholders.
- Strategic fit or execution risk: technology-integration complexity, cultural / operational mismatches noted by authors, synergy achievability.
- Any precedents or comparable deals that make this deal stand out as higher / lower risk.

**What NOT to include:**

- Generic deal risks that apply to all M&A (e.g., "regulatory approval required").
- Opinions or sentiment without a factual basis.
- Facts that only emerged after the first week post-announcement. Important: facts that were visible at announcement but only flagged later are fair game. Facts that were visible at announcement that could have foretold a later-emerging risk are ALSO fair game (e.g. product overlap that regulators flag months later is still a valid rubric item if the overlap was visible at announcement).

**Output format:** Produce a `<rubric>...</rubric>` block with 3–8 numbered items (fewer if genuinely fewer non-obvious catches exist; never pad with generic observations). Each item: 1–2 sentences. State the specific fact / insight and when it was knowable. Be precise and concise. Source each item from tools (do not rely on internal knowledge).

**Research approach:**

1. Perform an initial round of research using your tools to gather facts that were available at announcement.

---

2. Perform a thorough post-mortem analysis of the deal using all tools, focusing on identifying non-obvious insights that a sophisticated analyst should have caught from information available within the first week after announcement.

3. Based on your research, identify the most important 3–15 facts / insights that a sophisticated analyst should have caught from information available at announcement or within the first week after announcement. Remember to focus on non-obvious catches, not generic deal risks.

**Deal context:** {deal_context}

## D.2 Rubric Grading

Each model report is graded against its rubric by `gpt-5` using the prompt below. The grader assigns each rubric item one of three labels (Fully / Partially / Not Covered) and returns a coverage score in $[0, 1]$ together with a short explanation. The score reported as "rubric coverage" in Section 5.2 is the mean of this per-deal coverage score over the $> \$1B$ test set.

---

**Rubric Grader (`gpt-5`)**

You are an expert merger-arbitrage analyst. Score ONLY rubric coverage with partial credit.

**Scoring rules (semantic, not literal matching):**

- Identify the core claim for each rubric item (the decision-relevant substance). Treat extra clauses (exact dates, doc names) as supporting unless they change substance.
- Assign exactly one label per item:
  - **Fully Covered (1.0):** The model's report clearly states the core claim with correct polarity and material qualifiers. Paraphrases / synonyms and small numeric approximations are acceptable (e.g., "short-form merger after tender" ≈ "251(h) merger"; "~6–7%" ≈ "6.7%").
  - **Partially Covered (0.5):** The core idea appears but is missing critical qualifiers or is materially incomplete (e.g., mentions tender but omits majority tender / HSR gating; notes shareholder support without indicating irrevocable nature or magnitude).
  - **Not Covered (0.0):** The core claim is absent, contradicted, or only minor / supporting details are mentioned without the core claim.
- No double-counting across items.
- Compute `coverage_score` = (sum of item points) / total_items, rounded to two decimals.
- The source cited in the rubric is for context only; do not reference it in your scoring. The source for the model's report may be entirely different. Hence, do not expect the model to cite the same sources as the rubric (most likely, the model will identify it from primary sources like the filings, merger agreement, etc.).
- Your job is to evaluate the model's identification of the key facts from the rubric about the deal. If any rubric item is about a perception or opinion from a research paper, then do not use that for scoring (skip that item).

**Deal Context (for orientation):** {deal_desc}

**Rubric of Key Insights (items to check):** {rubric}

**Model's Report (to evaluate):** {detailed_report}

**Output format:** You may think and brainstorm inside a `<thoughts>...</thoughts>` block. Then return a JSON object:

- `score`: a float in $[0, 1]$, rounded to two decimals.
- `explanation`: a few sentences stating which items were fully and partially covered, and naming the most critical uncovered elements.

---

## D.3 Error Analysis

We take the 100 worst forecasting errors on the test set (largest squared-error samples) and classify each with `gpt-5` into one of five primary root causes: (1) MISSING, STALE, OR INCORRECT CONTEXT (the prompt was missing or stale on a **decision-relevant** fact), (2) MISSED SIGNAL key fact(s) were in the prompt but the model under-weighted it), (3) MISCALIBRATION (the model identified the right risks but assigned the wrong probability), (4) HALLUCINATION (the model introduced a fact not in the prompt and not true), and (5) STRUCTURAL LIMIT (the deal outcome was driven by an exogenous shock – e.g., a geopolitical shock or disease outbreak – that could not plausibly have anticipated.). Table 12 reports the resulting distribution.

| Primary root cause | N | % |
|---|---|---|
| MISSING, STALE, OR INCORRECT CONTEXT | 60 | 60.0 |
| MISSED SIGNAL PROVIDED IN THE PROMPT | 20 | 20.0 |
| MISCALIBRATION | 17 | 17.0 |
| STRUCTURAL LIMIT | 3 | 3.0 |
| HALLUCINATION | 0 | 0.0 |

*Table 12.* Primary root cause of the 100 worst forecasting errors made by GPT-4o-FT, classified by `gpt-5`.

**Missing or stale context examples.** To illustrate the texture of these gaps, Table 13 lists ten representative cases from the information-gap subset (N=60). For each case we report the model's predicted probability of a positive outcome, the realized outcome, and a description of the publicly available context that was missing or stale in the prompt at the sample date.

| Pred | Truth | Context missing or stale in the prompt |
|---|---|---|
| 0.98 | Failed | Missing the tender-offer terms (two-thirds minimum acceptance, offer price). |
| 0.95 | Failed | Missing the appraisal-payout cap that auto-terminates the merger if breached. |
| 0.86 | Failed | Missing that the tender price was set far below the prevailing market price. |
| 0.81 | Failed | Missing a blocking shareholder's >30% stake, which made acceptance implausible. |
| 0.75 | Failed | Missing the target board's public rejection of the proposal before the sample date. |
| 0.65 | Failed | Missing ISS and Glass Lewis recommendations against the deal. |
| 0.45 | Completed | Missing that both shareholder bodies had already overwhelmingly approved the merger pre-sample. |
| 0.20 | Completed | Stale 2019 lapsed bid in the prompt; the live 2025 recommended offer and its approvals were absent. |
| 0.19 | Completed | Missing that the offer had been declared unconditional once all regulatory conditions cleared. |
| 0.12 | Completed | Missing anchor-shareholder alignment on both sides and confirmation of key early regulatory approvals. |
| 0.07 | Completed | Missing the formal merger announcement on the sample date; the prompt framed the deal as an unsigned rumor. |

*Table 13.* Ten representative information-gap cases from the worst-100 set. In each case the model's reasoning is internally coherent given the prompt, but the prompt itself is missing a single decision-relevant fact that would have flipped the conclusion.

## D.4 Grounding Grader

To assess factual grounding, we run a two-stage `gpt-4.1` grader on each generated deal report. Stage 1 extracts a list of verifiable factual claims from the report. Stage 2 scores each claim against the source context provided to the forecaster on a 0–10 scale, normalized to $[0, 1]$. A claim is labelled GROUNDED if its normalized score is $\geq 0.8$, REASONABLY_INFERRED if $\geq 0.5$, UNCERTAIN if $\geq 0.2$, and CONFLICTING otherwise. The headline "unsupported-claim rate" reported in Section 5.2 is the share of CONFLICTING claims.

---

**Claim Extraction (`gpt-4.1`)**

You are an expert document analyst. Your task is to identify factual claims from an AI-generated financial forecast report that should be verified against source documents.

You will be given:

- `Context`: the source information provided to the forecaster.
- `Content`: the forecast report to analyze for claims.

**Extract claims that are:**

1. **Specific factual assertions** - dates, numbers, percentages, company names, deal terms.
2. **Attributed statements** - what authors said, what filings show, regulatory decisions.
3. **Time-sensitive claims** - status updates, timeline estimates, current conditions.

**Do NOT extract:**

- Generic forecasting methodology descriptions.
- The forecaster's own probability estimates or reasoning.
- Obvious restatements of provided context.
- Meta-descriptions ("This report analyzes…").

---

- Guesses or speculations that the model clarifies as coming from its own internal knowledge.
- Opinions, predictions, or subjective statements.
- Atemporal, common-sense knowledge.

For each claim, provide (i) the normalized claim text (self-contained, with enough context to verify) and (ii) an anchor substring from `Content` where the claim appears.

**Output:** JSON with two arrays — `claims` (list of extracted claim strings) and `anchor_substrings` (list of exact substrings from `Content` where each claim appears).

---

**Claim Grounding Scorer (`gpt-4.1`)**

You are a rigorous fact-checker. Your task is to assess how well a claim is supported by the provided context.

**Context is your ONLY source of truth.** Do not use external knowledge.

**Rating scale** (0.0 **to** 10.0)**:**
- **10.0:** claim is verbatim or trivially rephrased from context.
- **8.0-9.9:** clear, complete restatement with no altered details.
- **5.0-7.9:** reasonably inferred from context with small logical steps.
- **2.0-4.9:** partially supported but missing or unclear on key details.
- **0.0-1.9:** unsupported or contradicted by context.

**Be strict about:**
- Entity identity (correct company, ticker, person).
- Numerical values (even small differences matter).
- Dates and timelines.
- Direction of statements (increased vs. decreased, approved vs. pending).

**IMPORTANT:** Your job is not to grade the validity or correctness of the context. If the context itself is wrong or contains conflicting information, do NOT penalize the claim for that. Only assess whether the claim is supported by the context as given. E.g., if the context states "Company A acquired Company B for \$5B" and also states the opposite, then either claim is considered supported.

**Output JSON:**
- `rating`: float in $[0.0, 10.0]$.
- `reasoning`: max 30 words.
- `supporting_quote`: exact quote from context, or null.
- `contradicting_quote`: exact quote from context, or null.

