# OpenReview forum: "Global Merger-Arbitrage Forecasting with Language Models"
_ICML.cc/2026/Conference — ICML 2026 regular_

### Official Review · Reviewer_4QbW · 2026-02-16

**Soundness:** 3
**Presentation:** 2
**Significance:** 2
**Originality:** 1
**Overall Recommendation:** 4
**Confidence:** 3

**Summary:**

This paper proposes an LLM-based system for forecasting merger-arbitrage outcomes using domain-specific retrieval, specialized research agents, long-context reasoning and a finetuned forecasting model trained on hindsight-guided reasoning traces. The system produces probabilistic predictions over deal categorical deal outcomes (Succeed+, Fail+, Fail−), time-to-completion estimates, and explanatory reports. The pipeline is evaluated on a dataset of 1,648 M&A deals with temporally separated splits and Brier score metrics. The authors report that their method outperforms market-implied probabilities and an XGBoost baseline based on structured features. They also claim successful deployment of this system as a decision-support tool in a real investment workflow.

**Compliance With Llm Reviewing Policy:**

Affirmed.

**Final Justification:**

I increased my score from 2 to 4 given the authors rebuttal and the supplementary materials provided.
I have, however, lowered my confidence from 4 to 3 reflecting my lack of familiarity with ICML expectations for pure application-type papers.

**Key Questions For Authors:**

1. The proposed method relies on hindsight-guided reasoning traces as process supervision for finetuning. Have you compared this approach against outcome-only optimization (e.g., reward-based fine-tuning without explicit reasoning trace supervision, such as GRPO-style training)?

2. In the paper you state that the Murphy decomposition results suggest that frontier LLMs already exhibit strong discrimination but poor calibration, and that finetuning primarily improves calibration. Could you provide further experiments justifying the need for full finetuning of the forecasting model? Would simpler calibration method applied to the LLM's predictions not suffice (e.g. Platt scaling, isotonic regression)?

3. How do you ensure that hindsight-guided deal reports do not implicitly encode future information besides prompt constraints?

4. Will any portion of the dataset, retrieval corpus, or evaluation pipeline be released or reproducible with public or synthetic data?

**Limitations:**

The biggest limitation of this work is the dependency on proprietary and non-public data sources and lack of implementation details or access to code which introduces severe reproducibility concerns. I am willing to re-evaluate my score subject to authors addressing the reproducibility concern.

The paper also fails to provide a qualitative discussion of the reports generated by the pipeline or the failure modes of the system. Such insights may prove valuable for future research and deployment in high-state forecasting applications.

**Strengths And Weaknesses:**

### Strengths

- **Well-motivated application domain.** The focus on merger-arbitrage forecasting is compelling. The task is clearly articulated as requiring integration of complex legal, regulatory, and financial documents and ongoing event streams, which is a realistic and challenging setting for LLM-based forecasting.

- **A strong empirical scale.** The dataset spans 1,648 deals across multiple countries with temporally defined train/validation/test splits and over 4,000 forecast instances, which is a large and highly realistic setup.

- **Evaluation design and metrics.** The use of multiple weighted Brier scores is appropriate for the financial context and goes beyond standard accuracy metrics. The inclusion of Murphy decomposition to separate calibration and discrimination is also a methodological strength.

- **Good attempt at addressing temporal leakage.** The authors explicitly discuss leakage risks, enforce knowledge cutoffs, and restrict retrieval to timestamped documents prior to the forecast date, which shows awareness of known pitfalls in LLM forecasting evaluation.

- **Comprehensive ablations.** The ablation analysis (e.g., data scaling, oversampling, removal of research agents) provides useful insight into which components of the pipeline contribute to performance gains.


### Weaknesses

- **Possibly insufficient methodological novelty.** The core technical approach is largely an engineering integration of existing components: RAG-style retrieval, multi-agent analysis, ensembling, and finetuning of a frontier LLM. While the domain specialization is valuable, the paper seems not to introduce any technical novelties or practical insights generalizable to other domains.

- **Reproducibility concerns.** The absence of any appendix or supplementary material is a major issue given the complexity of the system. Critical details are missing, including: exact prompts and agent specifications, finetuning hyperparameters and dataset construction details, examples of hindsight supervisory reasoning chains. Moreover, the system relies on proprietary data sources (e.g., Bloomberg M&A data, broker research, internal document corpora), making independent replication effectively impossible. This substantially weakens the scientific value for the broader ML community. Publication of the code, with database placeholders, would significantly strengthen the contribution.

---

> ### Author Rebuttal · Authors · 2026-03-31
>
> We thank Reviewer 4QbW for their insightful and positive feedback. We are encouraged that the review identified many strengths in the paper (motivation, scale, evaluation design and metrics, control of temporal leakage, ablations). We now address the questions raised. The use of proprietary data and reproducibility is comprehensively addressed in the first rebuttal (to Reviewer jyQT).
>
> >Technical novelties and practical insights of the work
>
> Our technical novelty is a forecasting system and supervision recipe tailored to a specialist, high-stakes domain where the relevant evidence is long, technical, and heterogeneous - unlike prior LLM forecasting work that primarily uses short news context over mixed-topic question banks. Concretely, we contribute
>
> 1. A task setting that mitigates the shallow, generic retrieval and prompting and information leakage concerns that have characterized prior work using LLMs for judgmental forecasting (Sec 2, 3.3)
>
> 2. A two-stage agentic architecture that decomposes the problem into structured, domain-specific research and probabilistic synthesis, and
>
> 3. A hindsight-guided target construction scheme that leverages postmortems to generate temporally valid gold reasoning traces,and applies time-decayed label smoothing that blends realized outcomes with market-implied probabilities.
>
> **Practical insight.** We show that LLM-based forecasting can be effective in a specialist, economically consequential setting, producing actionable, citation-backed deal reports in production.
>
> > Have you compared this approach against outcome-only optimization?
>
> Yes, we report results with outcome-only supervision in Row 4 of the table in response to reviewer BPqS. Fine-tuning without hindsight-guided traces underperforms the full model. Furthermore, our preliminary experiments with GRPO-style outcome-only optimization were not effective. This is likely due to the limited back history available for training, strong class imbalance, and the sparse terminal reward for the long-context analysis.
>
> > Would simpler calibration method applied to the LLM's predictions not suffice (e.g. Platt scaling, isotonic regression)?
>
> In the table below, we demonstrate the performance of various post-hoc calibration methods: Platt Scaling, Isotonic Regression, and Temperature Scaling.
>
> **Only isotonic regression improves on the base model, and it still trails GPT-4o-FT.**
>
> Columns are as defined in the paper (Sec 3.4).
>
> | Model | Brier_B | Cal_B | Disc_B | Brier_S | Brier_$ | MAPE | MSD | $\rho$ |
> | --- | --- | --- | --- | --- | --- | --- | --- | --- |
> | Market $p^m$ | 0.229*** | 0.046 | 0.183 | 0.454*** | 0.319*** | -- | 0.000 | 1.000 |
> | XGBoost | 0.186** | **0.035** | 0.153 | 0.280*** | 0.246*** | -- | 0.064 | 0.753 |
> | GPT-4o | 0.201*** | 0.089 | 0.111 | 0.226** | 0.234*** | 0.463** | 0.133*** | 0.299 |
> | Ours (GPT-4o-FT) | **0.151** | 0.039 | 0.112 | **0.187** | **0.178** | **0.385** | 0.139 | 0.358 |
> | GPT-4o + Platt | 0.214*** | 0.101 | 0.111 | 0.247*** | 0.256*** | 0.463** | 0.148 | 0.302 |
> | GPT-4o + Isotonic | 0.186*** | 0.075 | **0.110** | 0.219*** | 0.228*** | 0.463** | 0.149 | 0.314 |
> | GPT-4o + Temp-Scale | 0.222*** | 0.109 | 0.111 | 0.254*** | 0.263*** | 0.463** | 0.149 | 0.300 |
>
> > How do you ensure that hindsight-guided deal reports do not implicitly encode future information besides prompt constraints?
>
> We agree this is a risk and mitigate it structurally. First, the teacher model only uses the postmortems to identify which pre-$t$ evidence is salient. Accordingly, the gold traces must cite each claim with evidence from the pre-$t$ documents provided. Second, we manually audit random samples and also run an independent LLM-based grounding grader. Under this check, the fine-tuned model shows no increase in unsupported content: GPT-4o-FT has a mean unsupported-claim rate of 0.1% while making 33 total claims per report on average, versus 0.3% for the base model while making 9 total claims per report.
>
> > The paper also fails to provide a qualitative discussion of the reports
>
> We appreciate the feedback and will include further qualitative analysis in the revised paper. Concretely, we score each report against a deal-specific checklist of key risks and mitigants visible on day 1. For example, for a deal like Lilly / Verve (2025), the rubric may contain tender mechanics and support agreements, limited antitrust risk, and CVR milestone design. We will also add a qualitative failure analysis: investigating the worst GPT-4o-FT errors shows that the primary failure point is missing context (e.g. absent news on shareholder approval or failed talks) rather than over/underconfidence on the model's part.
>
> In light of the many strengths highlighted and our responses addressing the concerns raised, we respectfully ask Reviewer 4QbW to reconsider the current reject recommendation. If the concerns raised have been addressed, an updated score would be greatly appreciated.

---

> > ### Author Rebuttal · Reviewer_4QbW · 2026-04-01
> >
> > My concerns regarding the underlying techniques used were resolved.
> > However, what remains is the concern regarding reproducibility.
> > The authors state:  will add an appendix with all prompts, example reports and reasoning chains and any other hyperparameters or details requested.
> > However, this is not part of their original submission, neither did the authors include a link to relevant Figures / Tables showing these details.

---

> > > ### Author Response · Authors · 2026-04-04
> > >
> > > Thank you for the update. We're glad that the concerns about the approach are resolved.
> > >
> > > We weren't sure about the protocol for providing an anonymized link. We provide it now, including hyperparameter details for finetuning the forecast LLM and XGBoost model, the full set of prompts used, an example final report, and an example reasoning chain.
> > >
> > > https://anonymous.4open.science/r/global-mna-forecasting-043A/

---

### Official Review · Reviewer_BPqS · 2026-03-07

**Soundness:** 2
**Presentation:** 2
**Significance:** 2
**Originality:** 3
**Overall Recommendation:** 4
**Confidence:** 4

**Summary:**

The paper builds an LLM-based forecasting system for M&A outcomes using retrieval-augmented research agents and a finetuned GPT-4o model. And it is evaluated on 400+ deals across 42 countries.

**Compliance With Llm Reviewing Policy:**

Affirmed.

**Final Justification:**

The new ablations directly address my three weaknesses. The isolation of the postmortem contribution (W2) is exactly the experiment I requested, BrierB regressing from 0.151 to 0.183 without postmortem guidance clearly demonstrates its value. The training-without-smoothing ablation (W3) similarly disentangles the market-implied signal from independent reasoning. On proprietary data (W1), the ICML precedent argument is fair, and the commitment to releasing prompts and reasoning chains is a meaningful step. My concerns are resolved and I am raising my score to 4.

**Key Questions For Authors:**

1. How much of the system could be reproduced using publicly available datasets?

2. Can the authors provide additional ablations that isolate the effect of: finetuning vs. no finetuning, hindsight-guided targets vs. standard outcome supervision, the research-agent pipeline vs. simpler retrieval baselines

3. Because the smoothed targets incorporate market-implied probabilities, have the authors tested training without this signal to evaluate whether performance improvements remain?

**Limitations:**

The authors acknowledge several limitations, including reliance on proprietary data sources and the challenges of preventing information leakage in forecasting systems. They also discuss the restricted scope of their evaluation to merger-arbitrage forecasting, which may limit generalization to other domains.

**Strengths And Weaknesses:**

Strengths:
The system is evaluated on a realistic merger-arbitrage forecasting task and it can be deployed as a decision-support tool in a financial setting. The ablation experiments suggest that different components of the system contribute to performance improvements.

Weakness 1: The system relies on a proprietary stack (Bloomberg M&A and others) that prevents independent verification or reproduction by the broader research community.

Weakness 2:  The paper cannot cleanly separate how much gain comes from (a) the 12 specialized research agents, (b) the hindsight-guided gold targets, and (c) finetuning. The ablations in Table 6 only ablate individual agents against the full finetuned system, they never test the gold target construction methodology in isolation.

Weakness 3: The smoothed training targets incorporate market-implied probabilities derived from stock prices. Because these prices reflect aggregated public information that may overlap with the textual context used by the model, there is a potential circularity risk where the model partially learns signals already embedded in market expectations. While the paper evaluates correlation with market probabilities, additional experiments training without this signal would help clarify how much predictive power comes from independent reasoning versus market-derived priors.

---

> ### Author Rebuttal · Authors · 2026-03-31
>
> We thank Reviewer BPqS for their time and feedback. A common concern around the use of proprietary data and reproducibility is comprehensively addressed in the first rebuttal (to Reviewer jyQT). We now address the remaining questions.
>
> > Separating the gain from (a) the 12 specialized research agents, (b) the hindsight-guided gold targets, and (c) finetuning. Can you test the gold target construction methodology in isolation?
>
> To address this, we add an ablation that holds the model and agent stack fixed and varies only the golden traces used for supervised fine-tuning. In Row 4 of the table below, we ablate the postmortem guidance, and the performance regresses: the balanced brier score increases from 0.151 to 0.183 (p < 0.01). These results indicate that SFT on post-hoc insight provides additional supervision signal.
>
> > The smoothed training targets incorporate market-implied probabilities derived from stock prices. Because these prices reflect aggregated public information that may overlap with the textual context used by the model, there is a potential circularity risk where the model partially learns signals already embedded in market expectations. While the paper evaluates correlation with market probabilities, additional experiments training without this signal would help clarify how much predictive power comes from independent reasoning versus market-derived priors.
>
> We disentangle two questions here.
>
> 1. **Use of market signals at inference (“piggybacking”)**. Deal spreads and stock prices can appear in the model context (L184–94). However, our system outperforms baselines that have the same access to these signals, and it exhibits substantially lower correlation with market-implied probabilities than those baselines (L423). Together, these results indicate that the system’s textual analysis is additive to, rather than merely a proxy for, market priors.
>
> 2. **Role of market-implied smoothing during training (label construction)**. Separately, we test whether the training-time smoothing itself is important by training a version of our system without smoothing outcomes using market-implied probabilities. As shown below, removing this component degrades performance: the balanced Brier score worsens from 0.151 to 0.181 (p < 0.01). This supports the motivation and discussion in L349 (right): smoothing provides a useful training signal, while the inference-time results above show the model is not simply regurgitating market expectations.
>
>
>
> Additional ablations. Stars indicate two-sided paired bootstrap significance of the difference relative to Ours (GPT-4o-FT) (*** p < 0.01, ** p < 0.05, * p < 0.10). Columns are as defined in the paper (Sec 3.4).
>
> | Model | Brier_B | Cal_B | Disc_B | Brier_S | Brier_$ | MAPE | MSD | $\rho$ |
> | --- | --- | --- | --- | --- | --- | --- | --- | --- |
> | GPT-4o (no fine-tuning) | 0.201*** | 0.089 | 0.111 | 0.226*** | 0.234*** | 0.463** | 0.133 | 0.299 |
> | GPT-4o: Fine-tuning on golden traces (postmortem + outcome + market probability) | **0.151** | **0.039** | 0.112 | **0.187** | **0.178** | **0.385**  | 0.139 | 0.358 |
> | GPT-4o: Golden traces (postmortem + outcome; no market probability) | 0.181*** | 0.081 | **0.100** | 0.210*** | 0.211*** | 0.390 | 0.130 | 0.321 |
> | GPT-4o: Golden traces (outcome + market probability; no postmortem) | 0.183*** | 0.062 | 0.121 | 0.231*** | 0.216*** | 0.397 | 0.136 | 0.367 |

---

> > ### Author Rebuttal · Reviewer_BPqS · 2026-04-01
> >
> > Thank you for the detailed rebuttal. The new ablations directly address my three weaknesses. The isolation of the postmortem contribution (W2) is exactly the experiment I requested, BrierB regressing from 0.151 to 0.183 without postmortem guidance clearly demonstrates its value. The training-without-smoothing ablation (W3) similarly disentangles the market-implied signal from independent reasoning. On proprietary data (W1), the ICML precedent argument is fair, and the commitment to releasing prompts and reasoning chains is a meaningful step. My concerns are resolved and I am raising my score to 4.

---

### Official Review · Reviewer_jyQT · 2026-03-13

**Soundness:** 2
**Presentation:** 3
**Significance:** 2
**Originality:** 2
**Overall Recommendation:** 3
**Confidence:** 2

**Summary:**

This paper introduces a specialized AI system designed for a highly lucrative, high-stakes financial sector: merger arbitrage. Instead of asking an LLM random trivia or broad geopolitical questions based on news headlines  the authors built a system to read massive, complex financial documents to predict whether corporate mergers will actually close, fail, or result in a higher bid.

**Compliance With Llm Reviewing Policy:**

Affirmed.

**Final Justification:**

The rebuttal is helpful. I reinforce my prior assessment.

**Key Questions For Authors:**

please see weakness

**Limitations:**

I couldn't found a separate limitation section per se so please see weakness.

**Strengths And Weaknesses:**

while I like this papers idea and specialized focus of M&A outcomes for merger arbitrage, I have a few concerns from the ML side:

1. The system was trained on "hindsight" reports written by a teacher AI that already knew how the deals ended. Even with strict instructions to only use past data, the teacher AI likely highlighted specific clues exactly because it knew they would cause the deal to fail later. This gives the model an unrealistic advantage and makes its performance look better than it actually is.

2. As far as I read, this paper use completely secret, proprietary data (Bloomberg databases and a massive internal document store) alongside closed-source models (GPT-4o, GPT-5). Because no outside researcher can access the data or the exact model setups, the scientific community cannot verify or build upon these claims.

3. another issue is that authors tried to prove that their complex, 12-agent system is necessary, but they tested it backwards. Instead of taking the full 12-agent system and removing one agent at a time to see if the system broke down, they started with a basic setup and just added single agents one by one. This fails to prove that all 12 agents are actually needed when put together as many of them might just be repeating each other's work.

4. a minor issue as a non-expert in finance domain, I found the claim where the AI beats market pricing a bit unrealistic using a "P&L-weighted" score, as this score ignores the messy reality of trading. It completely leaves out transaction fees, the high cost of borrowing shares to short, and liquidity constraints (whether you can actually buy the stock at that price). Beating a simplified math formula on paper does not prove the system works in the real world.

Given these concerns I'm giving this paper a weak reject however I want to stress that im not an expert in finance to extensively evaluate the financial side of the contributions of this paper. therefore i'm lowing my confidence to 2.

---

> ### Author Rebuttal · Authors · 2026-03-31
>
> We thank all three reviewers for their careful reading and constructive feedback. We are encouraged that reviewers found our focus on forecasting merger-arbitrage outcomes to be “realistic” (BPqQ) and “compelling” (4QbW), with “comprehensive ablations” and that the scale of the evaluation and evaluation design was appreciated, along with our careful mitigation of temporal leakage.
>
> ### The use of proprietary data and reproducibility
>
> All three reviewers raised questions about reproducibility and proprietary data.
>
> - **Why proprietary data is unavoidable:**
>
>   - The core contribution depends on an information environment that resembles professional investing.
>   - That environment is inherently built from a **broad mix of proprietary and subscription sources** (i.e., “secret, proprietary data”, jyQT).
> - **Why this data cannot be released:**
>
>   - Only a small number of organizations can assemble this breadth of sources in the first place.
>   - Those sources are governed by licensing and contractual restrictions.
>   - We **do not have permission to redistribute** the underlying data, and we cannot change that.
> - **Why research on public data is not a meaningful substitute**
>
>   - Public datasets do not reproduce the **information set**, **constraints**, or **decision context** faced by professional investors.
>   - Using only public data would therefore turn the study into a **toy setting**, undermining the research.
> - **Our guiding principle: some openness is better than none:**
>
>   - Even though we cannot provide full replication (data/code), we are committed to maximizing reproducibility: we will add an appendix with all prompts, example reports and reasoning chains and any other hyperparameters or details requested by reviewers (4QbW).
> - **Context: proprietary / restricted data is within ICML norms**
>
>   - ICML author/reviewer instructions do not prohibit proprietary or restricted-access datasets.
>   - Recent ICML outstanding paper award recipients have relied on proprietary data and did not release datasets, code, or model weights, e.g.: Genie (ICML 2024), VideoPoet (ICML 2024), Scaling Rectified Flow Transformers… (ICML 2024), The Value of Prediction… (ICML 2025).
>   - As with these papers, our contribution is the methodological and empirical insight, demonstrated through rigorous experiments - not a dataset or training artifact.
>
> ### Other concerns
>
> > Training on “hindsight” reports gives the model an unrealistic advantage and makes its performance look better than it actually is.
>
> We are also very concerned about information leakage (Sec 3.3), but this claim is incorrect. The teacher model only sees the postmortems, outcomes, or future evidence for the train set and **not the test set**. Hence, hindsight supervision on the student model cannot inflate test performance (L382).
>
> > Agents in the system should be ablated one at a time from the full system, rather than cumulatively adding agents one-by-one.
>
> Our original ablations demonstrate that our specialist research stack - designed with merger-arb specialists - outperforms a simpler context construction. As requested, we ran additional leave-one-out ablations starting from the full GPT-4o-FT system. Brier scores are worse, though statistical significance is limited. We agree that some agents likely produce partially overlapping information. However, in contrast to prior work that systematically ignores domain-specific data, tools, and workflows (L64, right), our stack is intentionally engineered around expert workflows to maximize forecast quality and produce reports that are useful to human analysts.
>
> Stars indicate two-sided paired bootstrap significance of the difference relative to Ours (GPT-4o-FT) (*** p < 0.01, ** p < 0.05, * p < 0.10). Columns are as defined in the paper (Sec 3.4).
>
> | Model | Brier_B | Cal_B | Disc_B | Brier_S | Brier_$ | MAPE | MSD | $\rho$ |
> | --- | --- | --- | --- | --- | --- | --- | --- | --- |
> | GPT-4o | 0.201*** | 0.089 | **0.111** | 0.226** | 0.234*** | 0.463*** | 0.133 | 0.299 |
> | Ours (GPT-4o-FT) | **0.151** | 0.039 | 0.112 | **0.187** | **0.178** | **0.385 | 0.139 | 0.358 |
> | FT w/o Financing Risk | 0.155 | 0.037 | 0.118 | 0.206 | 0.194 | 0.405 | 0.137 | 0.373 |
> | FT w/o Filings | 0.156 | **0.036** | 0.120 | 0.190 | 0.193 | 0.404 | 0.136 | 0.366 |
> | FT w/o Stakeholder Ownership | 0.158 | 0.043 | 0.114 | 0.201 | 0.188 | 0.393 | 0.138 | 0.352 |
>
>
> > Minor issue: The "P&L-weighted" score leaves out transaction fees, stock borrowing, and liquidity constraints.
>
> Correct, we report Brier scores, not a trading backtest. In addition to modeling these costs, a backtest would require specifying trading rules and portfolio optimization, which is outside our focus on judgmental probabilistic forecasting. We use the Brier score because it is a proper scoring rule for binary-event probability forecasts, and we report multiple weighting schemes to address class imbalance (L181-8, right).

---

> > ### Author Rebuttal · Reviewer_jyQT · 2026-04-04
> >
> > Thank you for your detailed rebuttal. I retain my current assessment.

---

### Decision · Program_Chairs · 2026-04-30

**Decision:**

Accept (regular)

**Comment:**

This submission sits near the borderline, but I come out slightly on the positive side. Reviewers agreed that the application is compelling and realistic: the paper tackles a difficult, high-stakes forecasting problem that requires synthesizing long and heterogeneous financial documents, and the empirical study is substantial. The rebuttal was also effective. In particular, added ablations helped isolate the value of the hindsight-style postmortem supervision and disentangle the role of market-implied signals, which directly addressed several review concerns and led one reviewer to substantially improve their assessment.
The remaining reservations are real. The strongest concerns are reproducibility and evaluation transparency, because the system depends heavily on proprietary data and tooling, and some of the promised details appear to be intended for the final version rather than fully present in the current submission. There was also continued discussion about hindsight leakage in the teacher-generated reports and about whether the paper's primary contribution is methodological or mainly an impressive application system. Even so, the positive reviewers found the revised empirical case convincing, and the negative review was not driven by a technical flaw so much as by concerns about fit and evidence standards. On balance, I think the paper clears the bar as a low-priority acceptance, primarily on the strength of the realistic task formulation and the improved ablation evidence.